# β4GALT1 controls β1 integrin function to govern thrombopoiesis and hematopoietic stem cell homeostasis

Silvia Giannini[1]*, Melissa M. Lee-Sundlov[1,2], Leonardo Rivadeneyra [2], Christian A. Di Buduo[3], Robert Burns[2], Joseph T. Lau[4], Hervé Falet [1,2,5], Alessandra Balduini [3,6] & Karin M. Hoffmeister[1,2,7]*

Glycosylation is critical to megakaryocyte (MK) and thrombopoiesis in the context of gene mutations that affect sialylation and galactosylation. Here, we identify the conserved *B4galt1* gene as a critical regulator of thrombopoiesis in MKs. β4GalT1 deficiency increases the number of fully differentiated MKs. However, the resulting lack of glycosylation enhances β1 integrin signaling leading to dysplastic MKs with severely impaired demarcation system formation and thrombopoiesis. Platelets lacking β4GalT1 adhere avidly to β1 integrin ligands laminin, fibronectin, and collagen, while other platelet functions are normal. Impaired thrombopoiesis leads to increased plasma thrombopoietin (TPO) levels and perturbed hematopoietic stem cells (HSCs). Remarkably, β1 integrin deletion, specifically in MKs, restores thrombopoiesis. TPO and CXCL12 regulate β4GalT1 in the MK lineage. Thus, our findings establish a non-redundant role for β4GalT1 in the regulation of β1 integrin function and signaling during thrombopoiesis. Defective thrombopoiesis and lack of β4GalT1 further affect HSC homeostasis.

[1] Division of Hematology, Department of Medicine, Brigham and Women's Hospital, Harvard Medical School, Boston, MA, USA. [2] Translational Glycomics Center, Blood Research Institute, Versiti, Milwaukee, WI, USA. [3] Laboratory of Biochemistry, Biotechnology, and Advanced Diagnosis, Department of Molecular Medicine, University of Pavia, Pavia, Italy. [4] Department of Molecular and Cellular Biology, Roswell Park Cancer Institute, Buffalo, NY, USA. [5] Department of Cell Biology, Neurobiology, and Anatomy, Medical College of Wisconsin, Milwaukee, WI, USA. [6] Department of Biomedical Engineering, Tufts University, Medford, MA, USA. [7] Departments of Hematology and Biochemistry, Medical College of Wisconsin, Milwaukee, WI, USA. *email: sgiannini@plateletbiogenesis.com; khoffmeister@versiti.org

Significant efforts are focused on identifying the most suitable cellular and molecular targets to enhance platelet production after bone marrow (BM) transplantation or chemotherapy[1]. Megakaryocytes (MKs) reside in the BM and maintain the continuous production of circulating platelets in order to prevent bleeding. The underlying pathogenic mechanisms of low platelet count (thrombocytopenia) are categorized as: (1) defects in MK lineage commitment and differentiation; (2) defects in MK maturation; and (3) defect in platelet release (extension of cytoplasmic protrusions into the blood stream also termed as thrombopoiesis). Optimized platelet release depends on the localization of MKs at BM sinusoids and the organization of the MK demarcation membrane system (DMS), an essential precursor complex membranous structure for proplatelet formation before their release into the bloodstream[2]. The mechanisms of platelet release into the circulation remain elusive and are under intense investigation.

β1 and β3 integrins are presumed dispensable for thrombopoiesis because their deletion does not affect thrombopoiesis and platelet counts[3,4]. However, αIIbβ3 integrin function has to be tightly regulated to allow thrombopoiesis and platelet release. Outside-in signaling generated by a constitutively activated αIIbβ3 impairs thrombopoiesis in human MKs[5–8]. β1 integrin family members play a pivotal role in cell migration and adhesion, including MK progenitors (MKPs), suggesting a role for this large integrin family in thrombopoiesis[9–11]. The role of β1 integrin function and binding to extracellular matrix (ECM) ligands in thrombopoiesis is unclear.

According to the classical model of hematopoiesis, MKs derived from resident hematopoietic stem cells (HSCs) that undergo sequential differentiation through the MK-erythroid progenitor (PreMegE)[12] and MKP stages. Recent data show that a pool of platelet-primed HSCs is biased towards the generation of MKs, and this particular pool increases with aging and inflammation[13–16]. Recent data suggest that MK-rich niches are highly enriched in platelet-biased HSCs, whereas non-biased HSCs reside within other BM spatial compartments[17].

Genetic alterations responsible for sialic acid and galactose (Gal) metabolism in humans regulate circulating platelet count, specifically in patients with mutations in GALE and GNE[18–20]. In mice, sialylated derivatives of the glycan structure β1,4-N-acetyllactosamine (Galβ1,4-GlcNAc or type-2 LacNAc, hereafter referred to as LacNAc), regulate platelet lifespan, hepatic thrombopoietin (TPO) production, and thus HSCs and thrombopoiesis[21,22]. β-1,4-galactosyltransferase 1 (β4GalT1), one of the seven members of the β-1,4-galactosyltransferase family synthesizes LacNAc. β4GalT1 transfers Gal from UDP-Gal to terminal N-acetylglucosamine (GlcNAc) to regulate glycoprotein and glycolipid expression and function[23]. β4GalT1 generates glycan motifs critical for homing and migration of several cell types[24–26]. However, recent findings point to a more complex role for β4GalT1 in hematopoiesis. The promoter region of the evolutionarily highly conserved B4GALT1 gene encoding β4GalT1 is rich in enhancer sequences for transcription factors associated with thrombopoiesis, such as E2F1, cell identity regulatory programs, and hematopoietic tumor drug resistance[27,28]. B4GALT1 also plays a role as an enhancer gene in CRISPR/Cas9-expressing mouse models of acute myeloid leukemia (AML)[29].

Here we investigate the role of β4GalT1 in thrombopoiesis and HSC function using B4galt1[−/−] mice. Our data show that β4GalT1 is a crucial regulator of β1 integrin function in MKs, thrombopoiesis, and HSC homeostasis. Our data show (1) while MKs were fully differentiated and increased in number in the absence of β4GalT1, agalactosylation of β1 integrin rendered the integrin hyperactive leading to MK dysplasia with severely impaired DMS formation and thrombopoiesis and perturbed HSCs function and numbers. Remarkably, β1 integrin deletion in B4galt1[−/−] MKs restores thrombopoiesis; (2) consistent with β1 integrin hyperactivity, B4galt1[−/−] platelets adhere avidly to β1 integrin ligands laminin, collagen, and fibronectin but have normal platelet α-granule secretion and von Willebrand factor (vWF) binding; (3) TPO and CXCL12 regulate β4GalT1 to increase glycosylation of MK proteins. The results show that β4GalT1-dependent galactosylation regulates β1 integrin function during thrombopoiesis and leads to perturbed HSCs. Identification of common regulatory mechanisms that promote HSC function and thrombopoiesis should enable the rational design of therapies to regulate platelet output.

## Results

**Hemorrhage and perinatal mortality in B4galt1[−/−] mice.** Breeding heterozygous B4galt1[+/−] mice in a mixed 129S1/C57BL/6J genetic background yielded at embryonic day 14.5 (E14.5) homozygous B4galt1[−/−] mice at the expected Mendelian frequency of 24.3% (Table 1). At weaning, only 10.8% of mice were B4galt1[−/−]. Thus, about 55% of B4galt1[−/−] mice died before weaning, most of them right after birth. Mice bred in the same conditions in the C57BL/6J background had no surviving B4galt1[−/−] neonates or adults. B4galt1[−/−] fetuses at E14.5 were slightly smaller, and had pallor and hemorrhage, including bleeding into the pericardial cavity (Fig. 1a, b). B4galt1[−/−] surviving neonates were smaller than littermates (Fig. 1c), as previously reported[30], and mice that survived to adulthood remained smaller in size. Splenomegaly persisted until adulthood (Table 2), with no significant anemia during embryonic stages and adulthood (Fig. 1d and Table 2). At E14.5 total white blood cell (WBC) count was normal. By contrast, adult B4galt1[−/−] mice had a significant leukocytosis (high WBC count) (Fig. 1e), as previously reported[31]. Differential blood count revealed that the leukocytosis was mainly due to increased neutrophil numbers (Table 2), suggesting a decreased adhesion of neutrophils to endothelial selectins[31]. Severe thrombocytopenia (low platelet count) was observed during fetal stages and in adulthood (Fig. 1f, g and Table 2). Consistent with the severe thrombocytopenia, evidence supports hemorrhage at embryonic stage E14.5–E15.5 in B4galt1[−/−] fetuses. However, whether bleeding is the leading cause of βGalT1[−/−] mouse mortality at this stage remains unclear.

**The absence of β4GalT1 impairs thrombopoiesis and HSCs.** The changes in peripheral blood count led us to investigate the hematopoietic compartments of B4galt1[−/−] mice. Phenotypic

**Table 1 Percentage of mice born from B4galt1[+/−] mice.**

| Age | B4galt1[+/+] (%) | B4galt1[+/−] (%) | B4galt1[−/−] (%) |
|---|---|---|---|
| E14.5 (129S1/C57BL/6J) | 25.4 | 50.3 | 24.3 |
| Weaning (129S1/C57BL/6J) | 31.0 | 58.2 | 10.8 |
| Weaning (C57BL/6J) | 61.7 | 38.3 | 0 |

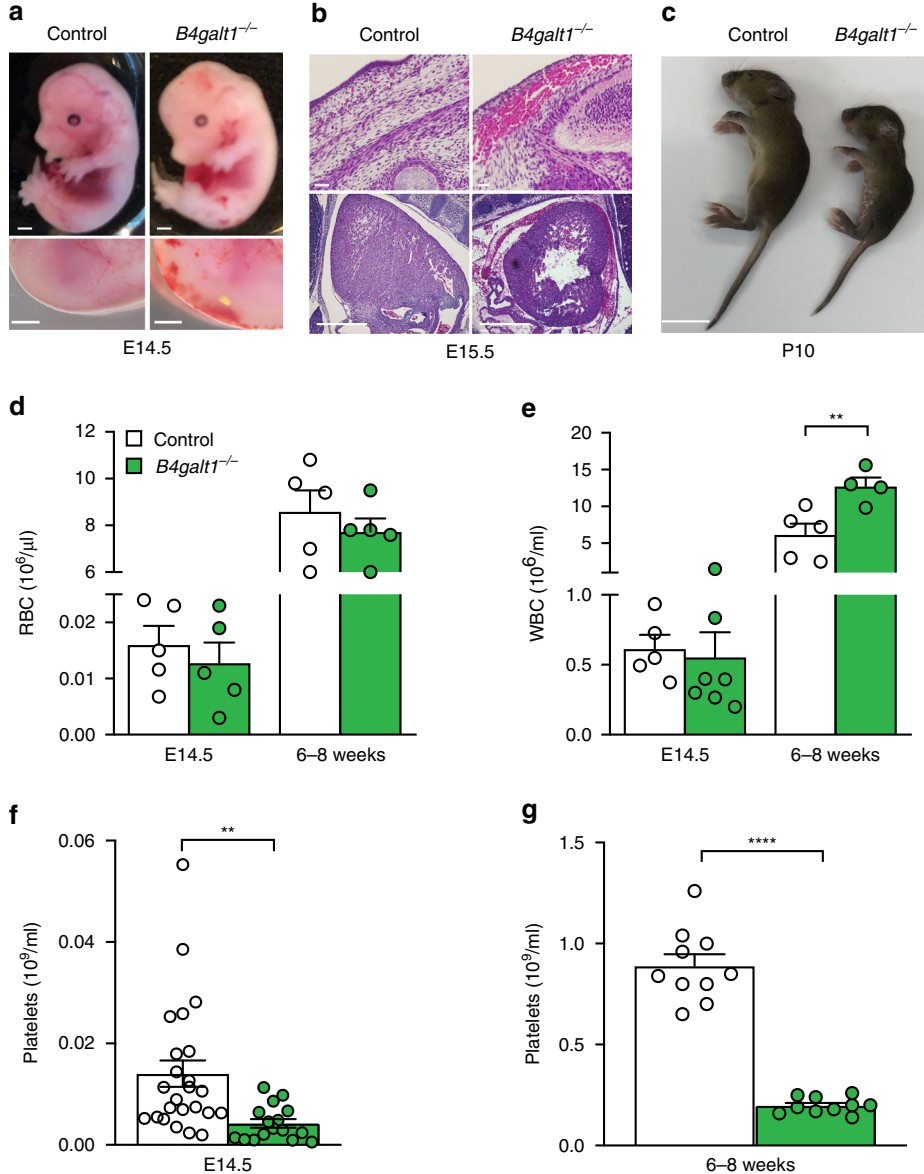

**Fig. 1 Hemorrhage and severe thrombocytopenia in *B4galt1*⁻/⁻ mice. a** Images of E14.5 control and *B4galt1*⁻/⁻ fetuses (scale bar: 1 mm). The magnification shows the dorsal side of the fetus. **b** Hematoxylin and eosin staining of sections of E15.5 control and *B4galt1*⁻/⁻ fetuses showing subcutaneous and intra pericardial cavity bleeding in *B4galt1*⁻/⁻ fetuses (scale bar: upper panel 20 μm, lower panel 500 μm). **c** Images of 10-day-old (P10) control and *B4galt1*⁻/⁻ mice (scale bar: 1 cm). Peripheral red (*n* = 5 in each group) **d**, white blood cell (*n* ≥ 5 per group) **e**, and platelet **f**–**g** counts (*n* ≥ 10 per group) as measured in control (white) and *B4galt1*⁻/⁻ (green) E14.5 fetuses and adult mice (6–8 weeks old). Data are expressed as mean ± SEM. Groups were compared using an unpaired Student's *t*-test, **\*\****p* < 0.01, **\*\*\*\****p* < 0.0001.

analysis of *B4galt1*⁻/⁻ BM showed normal percentage of B-cell (B220⁺), slightly decreased percentage of T-cell (CD3⁺) (*p* < 0.05), an increased percentage of CD11b⁺ cells (*p* < 0.05) (Fig. 2a). The percentage of red blood cells and their progenitors (TER-119⁺) were indistinguishable from control. Consistent with previous findings, the data suggest that the increase in the neutrophil count was due to neutrophil egress into the bloodstream in the absence of β4GalT1[31]. Further immunophenotypic analysis using signaling lymphocyte activation molecule (SLAM) markers[32] revealed a significantly increased percentage of long-term HSCs (LT-HSCs) (~2.5 fold) and multipotent progenitors (MPPs) (~3 fold) in *B4galt1*⁻/⁻ BM compared to control (Fig. 2b). The three-fold increased short-term HSCs (ST-HSCs) did not reach statistical significance. The distribution of common myeloid progenitors (CMPs) and granulocyte-macrophage progenitors (GMPs) was normal (Fig. 2c). By contrast, PreMegE,

MKPs, and differentiated MKs, defined as CD42b⁺ CD41⁺, were significantly increased in *B4galt1*⁻/⁻ BMs (Fig. 2c, d, Table 2). *B4galt1*⁻/⁻ BM colony-forming units were higher compared to control (Fig. 2e). The increase in HSC numbers led us to investigate their phenotypic and functional features further. While quiescence markers in *B4galt1*⁻/⁻ LT-HSCs were indistinguishable relative to controls (Fig. 2f), LT-HSCs expressed more of the platelet marker CD41 on their surface (Fig. 2g) supporting a highly expanded CD41⁺ subset of LT-HSCs. We measured LT-HSCs, ST-HSCs, and MPPs in the spleen of control and *B4galt1*⁻/⁻ mice. Similar to the BM data, LT-HSCs, and MPPs were increased (not significant) in *B4galt1*⁻/⁻ spleens compared to control, whereas there was no measurable difference in ST-HSCs (Table 2). The gating strategy utilized to identify LT-HSCs, ST-HSCs, MPPs, and CD41⁺ subset of LT-HSCs is shown in Fig. 2h.

**Table 2 Hematologic profiling of blood and bone marrows in WT, $B4galt1^{-/-}$, $Itgb1^{PF4+}$, and $B4galt1^{-/-}Itgb1^{PF4+}$ mice.**

| Parameter | WT | $B4galt1^{-/-}$ | $Itgb1^{PF4+}$ | $B4galt1^{-/-}Itgb1^{PF4+}$ |
|---|---|---|---|---|
| *Peripheral blood cells (6–8 weeks)* | | | | |
| RBCs ($10^6$/µl) | 8.6 ± 0.9 | 7.7 ± 0.5 | 8.1 ± 0.5 | 6.5 ± 0.6* |
| Neutrophils (K/µl) | 0.9 ± 0.1 | 4.0 ± 0.7*** | 0.6 ± 0.1 | 3.5 ± 1.0** |
| Lymphocytes (K/µl) | 8.6 ± 0.8 | 10.2 ± 1.3 | 7.4 ± 0.8 | 11.4 ± 2.7 |
| Monocytes (K/µl) | 1.5 ± 0.2 | 2.5 ± 0.5 | 1.7 ± 0.3 | 3.5 ± 0.8* |
| Platelets (K/µl) | 893 ± 36 | 199 ± 39*** | 990 ± 9 | 551 ± 29*** |
| Platelet half-life (h) | 63.2 ± 2 | 66.7 ± 5 | n.a. | n.a. |
| Plasma TPO (pg/ml) | 87.9 ± 5.5 | 328.5 ± 28.8*** | n.a | n.a |
| *Spleen* | | | | |
| mg/g | 3.4 ± 0.2 | 6.9 ± 1.1** | n.a. | n.a. |
| Megakaryocytes/field | 1.4 ± 0.2 | 4 ± 0.8** | n.a. | n.a. |
| LT-HSC (%) | 0.0018 ± 0.0007 | 0.006 ± 0.003 | n.a. | n.a. |
| ST-HSC | 0.003 ± 0.001 | 0.003 ± 0.001 | n.a. | n.a. |
| MPP | 0.03 ± 0.006 | 0.16 ± 0.09 | n.a. | n.a. |
| *Bone marrow megakaryocytes* | | | | |
| MK (% of live cells) | 1.5 ± 0.1 | 5.6 ± 1.4* | | |
| GPIbα (MFI) | 308.3 ± 24.6 | 244.4 ± 46.0 | n.a. | n.a. |
| GPIbβ (MFI) | 316.4 ± 5.1 | 319.0 ± 10.8 | n.a. | n.a. |
| GPIX (MFI) | 322.1 ± 10.2 | 326.8 ± 13.9 | n.a. | n.a. |
| αIIb (MFI) | 326.0 ± 46.2 | 298.5 ± 37.7 | n.a. | n.a. |
| β3 (MFI) | 145.1 ± 11.4 | 165.4 ± 30.1 | n.a. | n.a. |
| β1 (MFI) | 467.3 ± 34.6 | 581.2 ± 25.5* | n.a. | n.a. |
| CXCR4 (MFI) | 135.9 ± 12.05 | 159.2 ± 17.34 | n.a. | n.a. |
| *Primary transplant (% $GFP^+$ bone marrow cells)* | | | | |
| MK | 0.07 ± 0.002 | 0.4 ± 0.08* | n.a. | n.a. |
| CD3+ | 2.6 ± 0.3 | 0.8 ± 0.2** | n.a. | n.a. |
| B220+ | 11.6 ± 1.7 | 0.8 ± 0.2*** | n.a. | n.a. |
| CD11b+ | 80.2 ± 2.9 | 95.2 ± 0.8** | n.a. | n.a. |
| GPIbα (BM MKs, MFI) | 339.8 ± 31.2 | 317.7 ± 37.6 | n.a. | n.a. |
| β3 (BM MKs, MFI) | 160.3 ± 1.8 | 173.4 ± 2.6 | n.a. | n.a. |

$*p < 0.05$, $**p < 0.01$, $***p < 0.001$

While HSC platelet bias is TPO-dependent[14], increased inflammatory signaling induced by cytokines (TNF-α, IL-6, INF-α, TGF-β) also promotes expression of platelet markers and bias in HSCs[15]. Inflammatory cytokine levels in $B4galt1^{-/-}$ BMs were mostly unchanged except for a significant 50% decrease in TGF-β levels and an apparent, but non-significant increase in fibroblast growth factor 1 (FGF1) (Fig. 2i and supplementary Fig. 1), while other cytokines implicated in promoting platelet-biased HSCs (TPO, PDGF, TNF-α, IL-6, INF-γ) were indistinguishable relative to control. The four-fold higher plasma TPO levels (Table 2) compared to control, likely lead to an increase in the CD41+LT−HSC population but the combination of changes in cytokines may have additional effects on HSCs beyond platelet bias.

We next tested if the defect in thrombopoiesis is intrinsic to $B4galt1^{-/-}$ MKs by performing non-competitive transplants of donor BM cells expressing green fluorescent protein (GFP)[33] isolated from surviving mice into sub-lethally irradiated immunocompromised (NOD SCID gamma, NSG) recipient mice[34] (Fig. 3a). BM phenotypic analysis following transplant showed a pronounced increase in the myeloid lineage, which is promoted by irradiation-induced inflammation, and reduced $B4galt1^{-/-}$ B and T lineage reconstitution (Fig. 3b and Table 2). We measured increased MK numbers by flow cytometry (Fig. 3c), with no apparent difference in MK size and MK-specific receptor expression (Fig. 3d and Table 2). However, ~40% of $GFP^+$ $B4galt1^{-/-}$ MKs failed to localize at sinusoids compared to control MKs (38.6 ± 5.4% versus 63.7 ± 4.5%, respectively) (Fig. 3e). The $GFP^+$ $B4galt1^{-/-}$ platelet count remained profoundly decreased (<10% compared to control) throughout week 1–16 post-transplant (Fig. 3f). By contrast, transplanted $GFP^+$ control hematopoietic progenitor cells transplanted into NSG mice produced platelets normally, as evidenced by normal platelet counts. To further substantiate the intrinsic defect in thrombopoiesis, we performed transplants of donor fetal liver hematopoietic cells isolated from WT mice on the C57BL/J6 background expressing GFP[33] and non-fluorescent supporting fetal liver cells into lethally irradiated WT recipient mice (Fig. 3g). Similar to the data obtained in transplanted NSG mice, ~50% of $GFP^+$ $B4galt1^{-/-}$ MKs failed to localize at sinusoids compared to control MKs (31.2 ± 10.3% versus 62.6 ± 2.8%, respectively) (Fig. 3h). The $GFP^+$ $B4galt1^{-/-}$ platelet count remained profoundly decreased (<10% compared to control) throughout week 1–12 post-transplant (Fig. 3i). Hence, the defect in MK localization and thrombopoiesis is intrinsic to the $B4galt1^{-/-}$ MK and independent of environmental cues.

**$B4galt1^{-/-}$ MKs are intrinsically impaired.** The severe defect in thrombopoiesis in the absence of β4GalT1 led us to investigate thrombopoiesis in detail. During differentiation, MKs undergo endomitosis resulting in polyploidy, with MKs bearing one multilobulated nucleus. Granule formation and the development and organization of an extensive system of demarcation membranes that are the source of proplatelets and platelets are characteristic of cytoplasmic maturation[2]. Platelet formation and release are thought to be largely dependent on cytoskeletal elements[35], however the mechanisms of platelet formation and release remain unclear.

To distinguish whether the thrombocytopenia resulted from increased destruction of circulating platelets or from defective thrombopoiesis, we first measured the platelet half-life in adult $B4galt1^{-/-}$ and control mice. Surprisingly, $B4galt1^{-/-}$ platelet half-life was normal (Table 2). Thus, the severe $B4galt1^{-/-}$

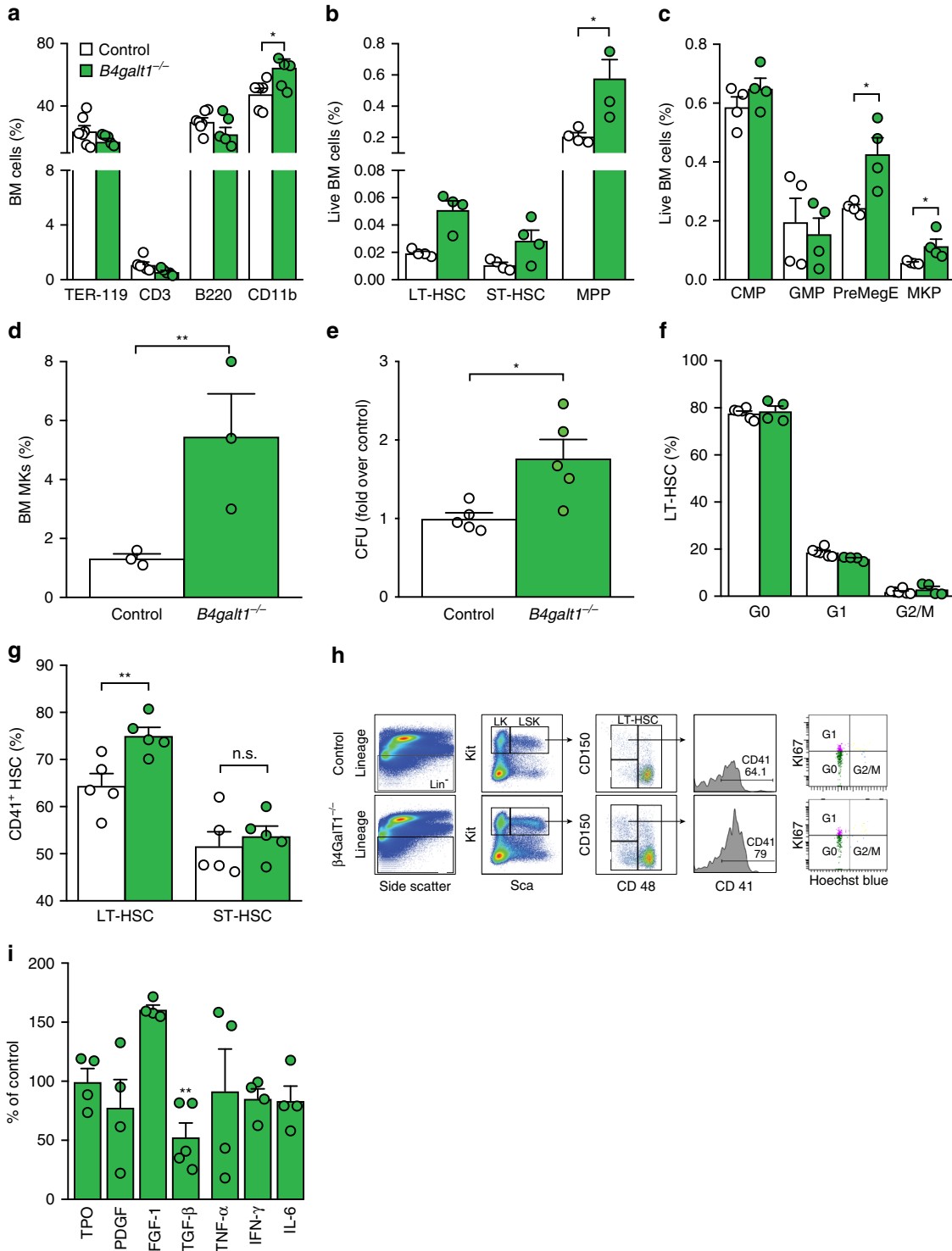

thrombocytopenia did not result from platelet destruction but reflected a primary failure in platelet formation. We next investigated localization, expression of specific MK markers, and ploidy in adult $B4galt1^{-/-}$ BM MKs compared with control MKs. MK numbers were increased in BM (~4-fold) (Fig. 2d and Table 2) and spleen (~3-fold) (Table 2) of adult $B4galt1^{-/-}$ mice relative to control. $B4galt1^{-/-}$ BM MKs appeared normal in size and had hyper lobulation of the nucleus as judged by immunofluorescence (Fig. 4a). To further characterize the maturation defect of $B4galt1^{-/-}$ MKs, BM cells were isolated, and MKs subjected to electron microscopy (Fig. 4b). Nuclei of

mutant MKs were hyper lobulated and appeared similar to control. While control MKs exhibited well-formed DMS, the recognized precursors of platelets, $B4galt1^{-/-}$ MKs had severely impaired DMS structures and lacked appropriate organization of the cytoplasm required for platelet formation.

To determine whether endomitosis was affected by the absence of β4GalT1, we performed a ploidy analysis of BM MKs. The modal ploidy class of $B4galt1^{-/-}$ MKs was 16N, although slightly reduced relative to control (Fig. 4c), indicating normal endomitosis of mutant cells. A fraction of mature $B4galt1^{-/-}$ MKs was not associated with sinusoids compared to control MKs (42.6% ±

**Fig. 2 Lack of β4GalT1 leads to a perturbed HSC phenotype.** Phenotypic analysis of **a** Ter-119[+], CD3[+], B220[+], and CD11b[+]. $n = 5$ in each group. Data are expressed as mean ± SEM. Groups were compared using an unpaired Student's $t$-test, $*p < 0.05$. **b** LT-HSCs, ST-HSCs, and MPPs ($n = 4$ in each group), and **c** CMPs, GMPs, PreMegEs, and MKPs cells ($n = 4$ in each group), and **d** MKs ($n = 3$ in each group) in BMs of control (white) and $B4galt1^{-/-}$ (green) mice. The molecular markers utilized to identify specific subpopulations of hematopoietic stem and progenitor cells are summarized in Supplementary Table 1. Data are expressed as mean ± SEM. Groups were compared using an unpaired Student's $t$-test, $*p < 0.05$; $**p < 0.01$. **e** Colony forming units obtained and quantified from BM HSCs isolated from control (white) and $B4galt1^{-/-}$ (green) mice BMs. $n = 5$ in each group. Data are expressed as mean ± SEM. Groups were compared using an unpaired Student's $t$-test, $*p < 0.05$. **f** Quantification of cells in G0, G1, and G2/M phase of the cell cycle within the LT-HSC compartment ($n = 4$ in each group), and **g** CD41[+] LT- and ST-HSCs ($n = 5$ in each group) in control (white) and $B4galt1^{-/-}$ (green) mice by flow cytometry. Data are expressed as mean ± SEM. Groups were compared using an unpaired Student's $t$-test, n.s. = not significant; $**p < 0.01$. **h** Flow cytometry plots showing the gating strategy to identify LT-HSC and the CD41[+] subset in control and $B4galt1^{-/-}$ BM cells. Lineage negative cells were plotted for Sca and Kit. The double positive population (LSK) was further plotted for CD150 and CD48. LT-HSCs were gated as CD150[+] and CD48[−] and evaluated for CD41 expression. **i** Quantification of TPO, PDGF, FGF-1, TGFβ1, TNFα, IFNβ, and IL-6 in the BM supernatant of control (white) and $B4galt1^{-/-}$ (green) mice. $n \geq 4$ per group. Data are expressed as mean ± SEM. Groups were compared using an unpaired Student's $t$-test, $**p < 0.01$.

5.7% versus 65.9 ± 3.7%, respectively ($n = 3$; $p < 0.05$)) (Fig. 4d) corroborating the impaired localization capacity of mature MKs at the sinusoidal level obtained after BM transplants (Fig. 3j). In vitro experiments investigating the ability of fetal liver-derived MKs to form proplatelets, the required cytoplasmic extensions that are precursors to platelets[35], revealed that $B4galt1^{-/-}$ MKs had a severe defect in proplatelet formation compared to controls (Fig. 4e, f). In summary, the ultrastructural analysis of $B4galt1^{-/-}$ MKs revealed a striking absence of platelet territories and cytoplasmic disorganization, resulting in defective thrombopoiesis.

**Increased activity of β1 integrin in $B4galt1^{-/-}$ MKs.** To identify β4GalT1 stage-specific requirements for platelet production, we examined the platelet levels of β1 tubulin, PF4, GPIb/IX, and β1 and β3 integrins, as specific markers of MK maturation[36]. Flow cytometry and immunoblotting analysis revealed regular expression of GPIbα, αIIbβ3, β1 tubulin, PF4 in $B4galt1^{-/-}$ BM MKs, compared to control (Fig. 5a, Supplementary Fig. 2), showing that β4GalT1 was not essential for their expression. Hence, the early and later stages of MK development proceeded unperturbed in the absence of β4GalT1. A significant increase in the surface and total β1 integrin subunit expression in $B4galt1^{-/-}$ MKs, but not platelets, was measured by flow cytometry and immunoblotting relative to control (Fig. 5b and Table 2).

Integrin-mediated adhesion requires binding of α and β subunits to a defined peptide sequence, which can be modulated by glycosylation of specific integrins[37–41]. In human and mouse β1 integrin glycosylation, changes explicitly in sialylation, regulate cell adhesion, and motility induced by cell differentiation or malignant cell transformation[27]. β4GalT1 synthesizes LacNAc, and the glycan structure is modified and capped by sialic acid decorations. Hence, we speculated that thrombopoiesis in $B4galt1^{-/-}$ mice is affected by impaired β1 integrin function and expression due to aberrant glycosylation. MKs express α2, α4, α5 and α6, and αIIb integrin subunits[9–11]. We tested the expression levels of α5 and α6 in MKs and platelets by immunoblotting of total cell lysates. MKs and platelets lacking β4GalT1 expressed α5 and α6 but both subunits had an apparent lower molecular weight compared to control (Fig. 5a). We next tested the expression of the platelet and megakaryocyte-specific αIIb subunit of the fibrinogen receptor αIIbβ3 by immunoblotting and detected a slightly lower apparent molecular weight in MKs and platelets compared with control (Fig. 5a). These data suggest that β4GalT1 adds Gal to α5, α6, and αIIb subunits.

Immunoblotting revealed that control MKs expressed the β1 integrin subunit as two glycoforms of 130 and 110 kDa, whereas control platelets predominantly expressed the 130 kDa glycoform (Fig. 5c). By contrast, $B4galt1^{-/-}$ MKs and platelets predominantly expressed the 110 kDa β1 subunit glycoform. Enzymatic removal

of N-linked glycans using PNGaseF reduced β1 integrin to the predicted molecular weight of ~95 kDa in both control and $B4galt1^{-/-}$ platelets (Fig. 5c). This data suggest that the apparent molecular weight of β1 integrin glycoforms in murine MKs likely results from differential glycosylation of the 12 putative highly conserved N-linked glycosylation sites[42]. To determine the expression of terminal Gal moieties by β1 integrin, we immunoprecipitated platelet β1 integrin from control and $B4galt1^{-/-}$ platelets and subjected the precipitates to SDS–PAGE and immunoblotting using monoclonal β1 integrin antibodies and ECL lectin. Lectin binding shows that β1 integrin does not express terminal Gal in the absence of β4GalT1 compared to control (Fig. 5d). These data show that circulating platelets express non-sialylated β1 integrin glycoforms with terminal Gal at steady-state and galactosylation of β1 integrin in MKs and platelets is β4GalT1-dependent.

We speculated that a lack of β4GalT1 affects β1 integrin signaling and function in MKs and platelets. Immunofluorescence of BM sections using an antibody that recognizes the activated form of β1 integrin (clone 9EG7) revealed increased expression of activated β1 integrin in $B4galt1^{-/-}$ MKs relative to control. Immunoblotting of unstimulated (resting) platelet lysates using a specific anti-phosphotyrosine (pTyr) antibody revealed three additional polypeptides of ~40, ~80, and ~125 kDa in $B4galt1^{-/-}$ platelets, compared to control (Fig. 5g). We hypothesized that the 125-kDa band was FAK, as integrin clustering at focal adhesions triggers FAK activation and phosphorylation. FAK is auto-phosphorylated at tyrosine residue 397 (Y397)[43], a signal that is further enhanced by the recruitment of the tyrosine kinase Src and subsequent phosphorylation of tyrosine residues 576 and 577 (Y576/577)[44]. Using specific antibodies directed against phosphorylated FAK residues Y397 and Y576/577, we identified the 125-kDa polypeptide in $B4galt1^{-/-}$ platelet lysates as FAK (Fig. 5h, i). The identity of the ~40 and 80-kDa polypeptides is unclear at this point. Together, the data show that β1 integrin signaling in the absence of β4GalT1 is hyperactive in MKs and platelets.

We next measured the function of platelets in vitro. We first tested the adhesion of platelets to immobilized fibrinogen (FG), collagen (COL), fibronectin (FN), and laminin (LN). Platelet adhesion to β1 integrin ligands FN, LN, and COL, was increased compared to control, while no difference was detected in platelet adhesion to immobilized FG, the αIIbβ3 ligand, or FG binding to platelets activated with the agonist thrombin, as measured by flow cytometry (Fig. 6a, b). vWF binding after botrocetin stimulation was similar to control (Fig. 6c). Furthermore, α-granule secretion, another primary function of platelets, was slightly but not significantly delayed following thrombin activation as determined by P-selectin exposure (an α-granule marker) on platelet surfaces (Fig. 6d). Hence, platelets had normal αIIbβ3 and vWF receptor

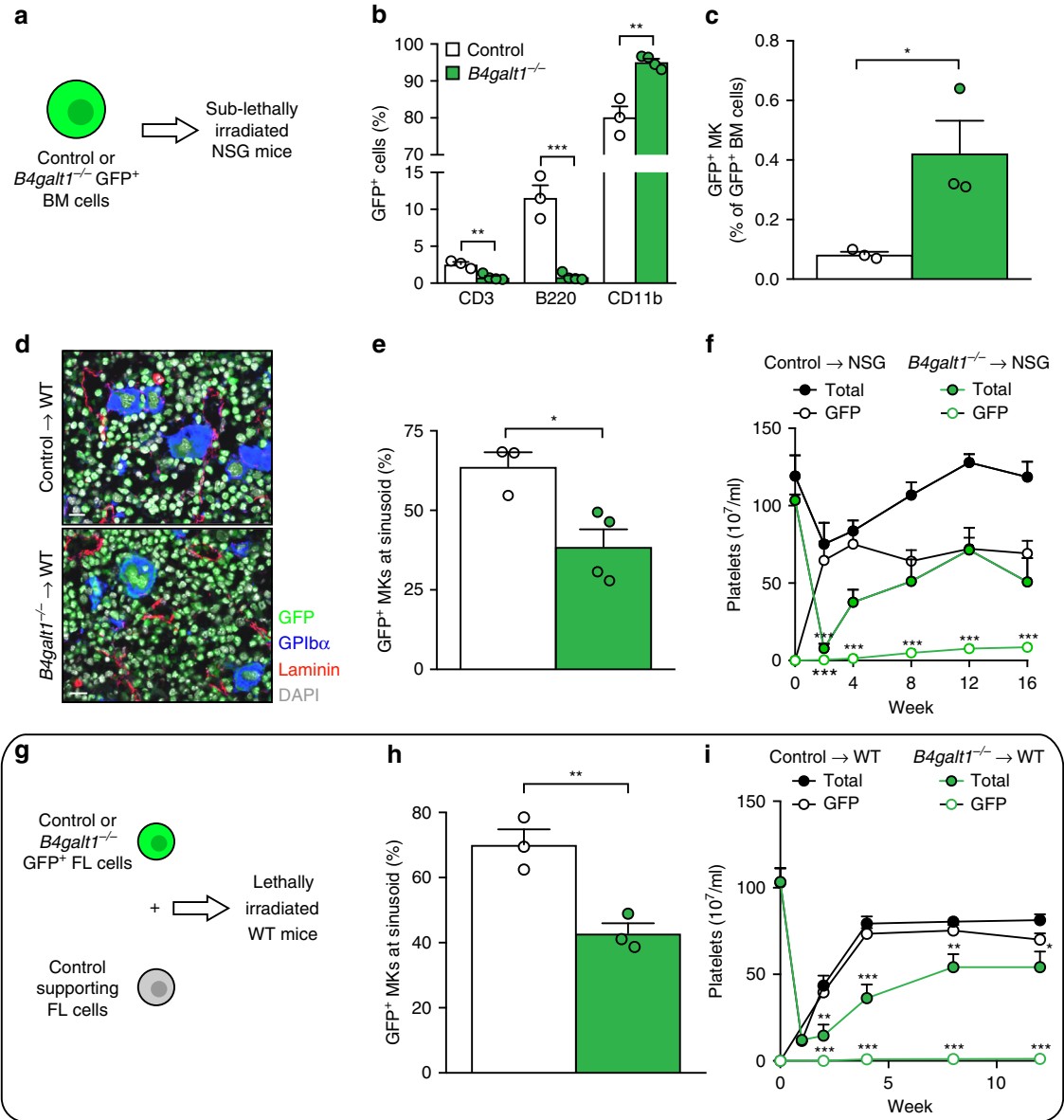

**Fig. 3 Intrinsic megakaryocytic abnormalities are retained following *B4galt1*⁻/⁻ HSC transplants. a** Scheme of bone marrow cell transplant; ubiquitin-GFP (GFP⁺) control or *B4galt1*⁻/⁻ BM cells were injected into sub-lethally irradiated NSG mice. **b** and **c** Quantification of GFP⁺ CD3⁺, B220⁺, CD11b⁺, and MKs BM cells in NSG mice transplanted with control (white) or *B4galt1*⁻/⁻ (green) BM cells. n ≥ 3 per group. Data are expressed as mean ± SEM. Groups were compared using an unpaired Student's *t*-test, \**p* < 0.05, \*\**p* < 0.01, \*\*\**p* < 0.001. **d** Immunofluorescence of BM sections of NSG mice following primary transplant with GFP⁺ control and *B4galt1*⁻/⁻-derived BM cells. BM sections were stained with anti-laminin antibody (red) to identify vessels, anti-GPIbα antibody (blue) to identify MKs, and DAPI (gray) to identify nuclei. Scale bar: 10 μm. **e** Quantification of GFP⁺ MKs at sinusoids in NSG mice following primary transplant with GFP⁺ control (white) and *B4galt1*⁻/⁻ (green) BM cells (n = 4 in each group. Data are expressed as mean ± SEM. Groups were compared using an unpaired Student's *t*-test, \**p* < 0.05. **f** Peripheral blood GFP⁺ platelets and total platelet count measured at 0, 4, 8, 12, and 16 weeks following transplants of control (black) and *B4galt1*⁻/⁻ (green) GFP⁺ BM cells into NSG mice. n = 9; in each group. **g** Scheme of fetal liver cell transplant; ubiquitin-GFP (GFP⁺) control or *B4galt1*⁻/⁻ fetal liver (FL) cells were isolated at E14.5 from C57BL/6 fetuses and injected into WT (C57BL/6) mice. **h** Quantification of GFP⁺ MKs at sinusoids in WT mice following transplant with GFP⁺ control (white) and *B4galt1*⁻/⁻ (green) FL cells. n ≥ 4 in each group. Data are expressed as mean ± SEM. Groups were compared using an unpaired Student's *t*-test, \*\**p* < 0.01. **i** Peripheral blood GFP⁺ platelets and total platelet count measured at 0, 4, 8, 12 weeks following transplants of control (black) and *B4galt1*⁻/⁻ (green) GFP⁺ FL cells into WT mice (n = 9). Data are expressed as mean ± SEM.

(GPIbα/β/V/IX) function but adhered avidly to β1 integrin ligands, supporting the notion that β1 integrin is specifically hyperactive in the absence of β4GalT1.

**B4galt1⁻/⁻ MKs lacking the β1 integrin have normal thrombopoiesis.** β1 and β3 integrins are presumed dispensable for

thrombopoiesis, an assertion based on the fact that their deletion does not affect thrombopoiesis and platelet counts[3,4]. By contrast, gain-of-function mutations in the genes encoding for αIIb and β3 result in defective DMS formation and thrombopoiesis thrombocytopenia, pointing to defective thrombopoiesis[5,6]. These data led us to hypothesize that hyperactive β1 integrin negatively affects DMS formation and thrombopoiesis. We reasoned that the

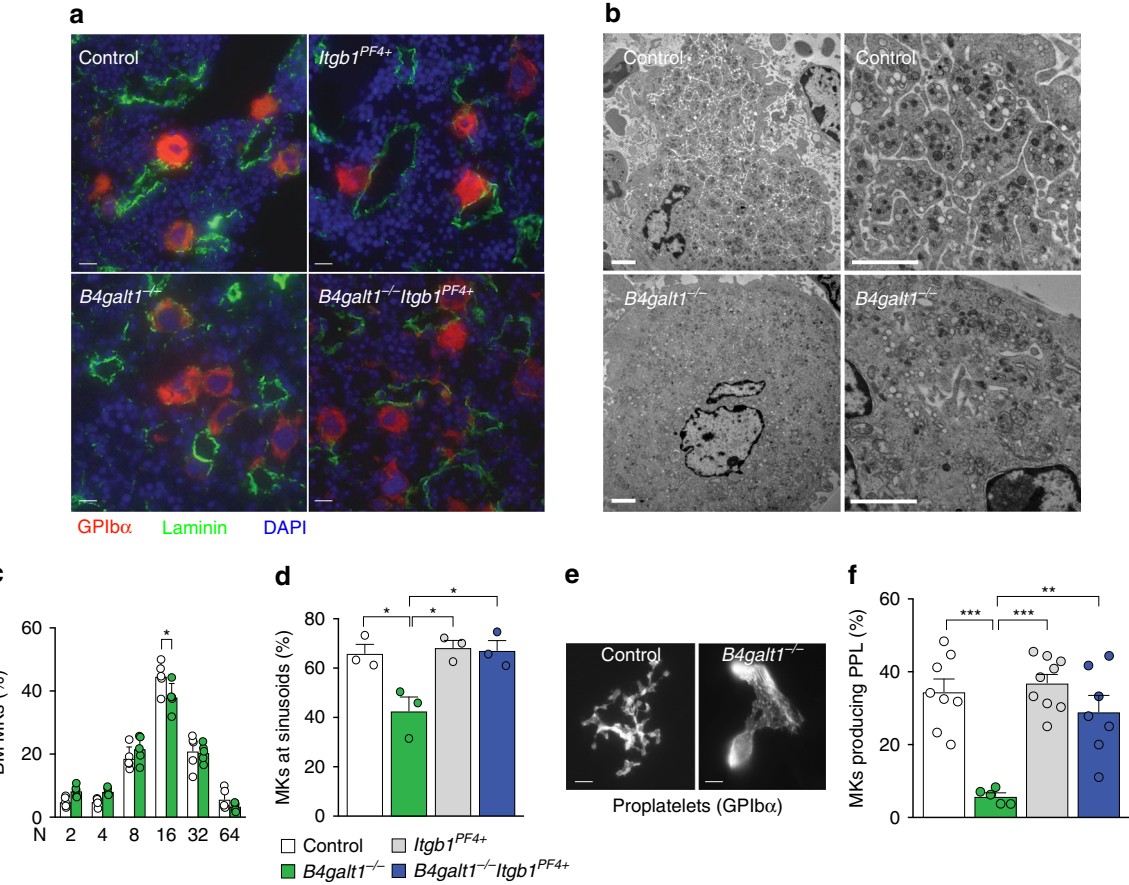

**Fig. 4 Lack of β4GalT1 inhibits MK maturation at the level of DMS and platelet formation. a** Immunofluorescence of control, $B4galt1^{-/-}$, $Itgb1^{PF4+}$, and $B4galt1^{-/-}Itgb1^{PF4+}$ BM sections stained for GPIbα (MKs, red), laminin (vessels, green), and DAPI (nuclei, blue). Scale bar: 10 μm. **b** Transmission electron microscopy images of MKs flushed from control and $B4galt1^{-/-}$ BMs (scale bar: 2 μm). **c** Flow cytometric quantification of BM MK ploidy in control (white) and $B4galt1^{-/-}$ (green) mice. $n = 5$ in each group. Data are expressed as mean ± SEM. Groups were compared using an unpaired Student's $t$-test, $*p < 0.05$. **d** Quantification of MKs at sinusoids (%) in the BM of control, $B4galt1^{-/-}$, $Itgb1^{PF4+}$, and $B4galt1^{-/-}Itgb1^{PF4+}$ mice. $n = 3$ in each group. Data are expressed as mean ± SEM. One-way ANOVA was used to compare each group, $*p < 0.05$. **e** Representative immunofluorescence image of control and $B4galt1^{-/-}$ fetal liver-derived MKs producing proplatelets stained for GPIbα (scale bar: 5 μm). **f** Quantification of cultured fetal liver cell-derived control, $B4galt1^{-/-}$, $Itgb1^{PF4+}$, and $B4galt1^{-/-}Itgb1^{PF4+}$ MKs producing proplatelets (%). $n \geq 5$ in each group Data are expressed as mean ± SEM. One-way ANOVA was used to compare each group, $**p < 0.01$; $***p < 0.001$.

deletion of the hyperactive form of the β1 integrin subunit in $B4galt1^{-/-}$ MKs would improve thrombopoiesis in $B4galt1^{-/-}$ mice. We, therefore, crossed $B4galt1^{-/-}$ mice with mice lacking β1 integrin specifically in MKs and platelets ($B4galt1^{-/-}/Itgb1^{fl/fl}$ PF4-Cre mice, hereafter referred to as $B4galt1^{-/-}Itgb1^{PF4+}$). Remarkably, deletion of β1 integrin in $B4galt1^{-/-}$ MKs restored: (1) MK localization at sinusoids (Fig. 4a, d); (2) proplatelet production in vitro (Fig. 4f); (3) integrin signaling in platelets (Fig. 5g–i); and (4) significantly improved circulating platelet count to 70% of control (Table 2). Other blood cell counts were not affected by β1 integrin deletion, except for a slightly decreased RBC count (Table 2). Together, the data suggest that glycosylation of β1 integrin affects ECM binding to regulate thrombopoiesis.

**LacNAc synthesis depends on β4GalT1 in HSCs and MKs.** Out of seven identified β4GalTs, β4GalT1 and β4GalT2 are the most efficient in synthesizing type-2 LacNAc[45]. The severe defects in $B4galt1^{-/-}$ HSC function and thrombopoiesis led us to evaluate the level of terminal βGal and enzymatic β4GalT activity in the hematopoietic cells of control and $B4galt1^{-/-}$ mice. Illumina mRNAseq expression analysis in control MKs, platelets (Plt), Lineage⁻ Sca⁺ c-Kit⁺ (LSK), dendritic cells (DC), and T cells

(TC) showed significantly higher expression of $B4galt1$ in platelets (adjusted $p < 0.001$) compared to other cell types, while $B4galt2$ mRNA expression remained very low in all cell types (<1 FPKM) (Fig. 7a). Platelets had a ~3.5-fold increased $B4galt1$ mRNA expression compared to MKs, suggesting an upregulation of $B4galt1$ mRNA during the final stages of MK maturation (Fig. 7a). The ability of LacNAc synthesis by MKs, platelets, and CD45⁺ cells was determined by measuring GalT enzymatic activity towards the acceptor benzyl-β-N-Acetyl-D-Glucosamine (βGlcNAc) in the presence of UDP-[³H]Gal substrate (Fig. 7b). $B4galt1^{-/-}$ MKs and platelets did not transfer measurable Gal onto the acceptor, compared to control MKs. By comparison, $B4galt1^{-/-}$ CD45⁺ cells had low but measurable enzymatic β4GalT activity relative to control.

Fluorescently labeled lectins *Erythrina cristagalli* lectin (ECL) and succinyl-Wheat Germ agglutinin (s-WGA) were used to determine the expression of the terminal βGal and βGlcNAc, respectively, in control and $B4galt1^{-/-}$ MKs, platelets, LSK cells, and CD45⁺ BM cells by flow cytometry. Figure 6c shows a typical structure of an N-linked glycan containing LacNAc and lectin binding. $B4galt1^{-/-}$ MKs, platelets, and LSK cells had significantly reduced terminal Gal and increased βGlcNAc expression compared to control (Fig. 7d, e). While its canonical specificity is

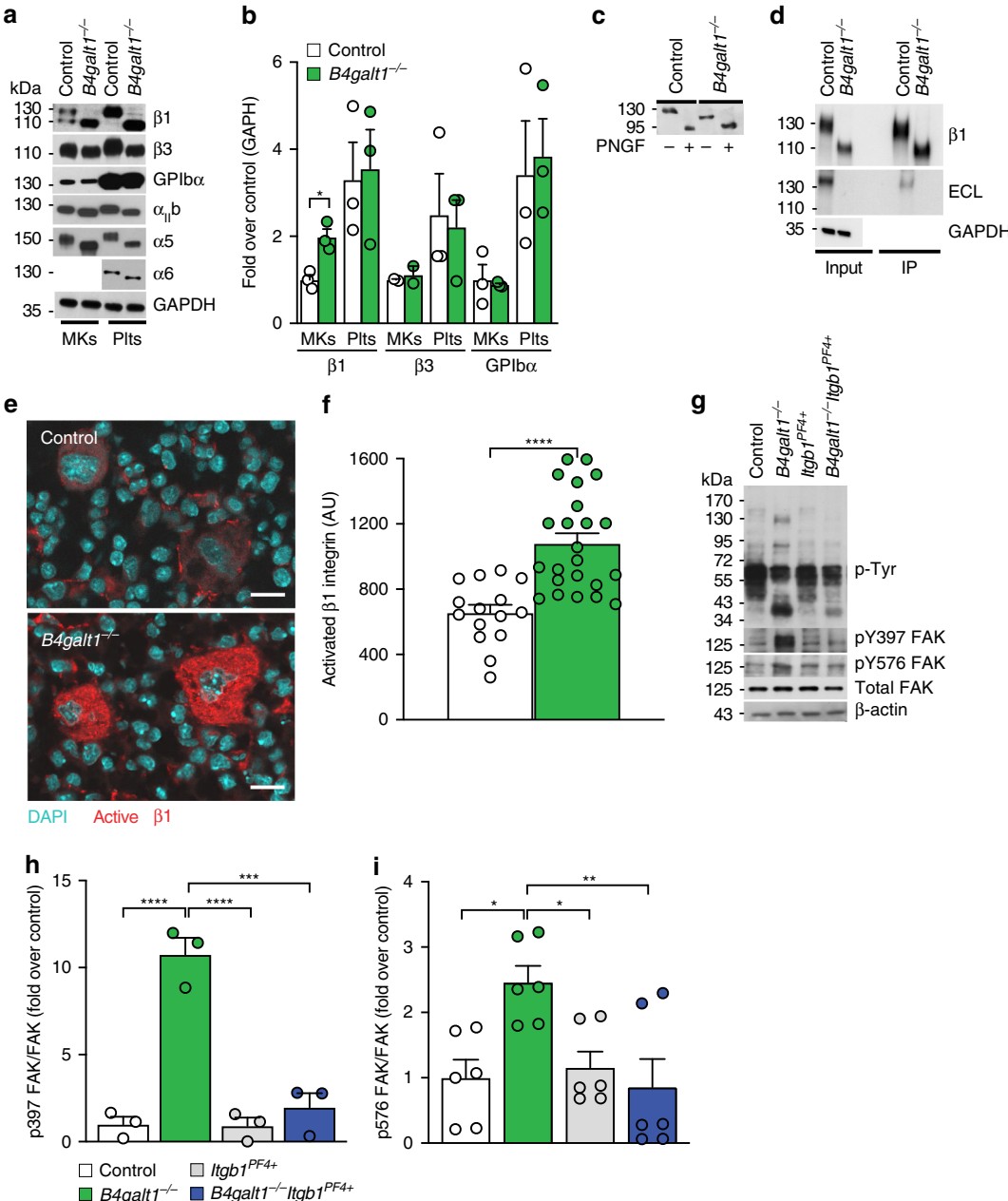

**Fig. 5 Deletion of hyperactive β1 integrin restores thrombopoiesis by _B4galt1_⁻/⁻ MKs. a** Total BM-derived MK and blood platelet lysates obtained from control and _B4galt1_⁻/⁻ mice were subjected to SDS–PAGE. Immunoblots were probed with mAbs directed against β1 and β3 integrin subunits, GPIbα, αIIb, α5, α6 integrin subunits, and GAPDH as loading control. **b** Densitometric quantification of immunoblots probed with antibodies directed against the β1 and β3 integrin subunits and GPIbα of control (white) and _B4galt1_⁻/⁻ (green) MKs and platelets. $n = 3$ in each group. Data are expressed as mean ± SEM. Groups were compared using an unpaired Student's _t_-test, *$p < 0.05$. **c** Control and _B4galt1_⁻/⁻ platelet lysates, treated (+) or not (−) with PNGaseF (PNGF) to remove all N-linked glycans were subjected to SDS–PAGE and immunoblots were probed with an anti-β1 integrin antibody. **d** Total platelet lysates (input) and platelet lysates immunoprecipated with anti-β1 integrin antibody (IP) obtained from control and _B4galt1_⁻/⁻ mice were subjected to SDS–PAGE. Immunoblots were probed with mAbs directed against β1 integrin subunit, ECL, and GAPDH. **e** Immunofluorescence of control and _B4galt1_⁻/⁻ mouse BM sections stained with the antibody 9EG7 directed against activated β1 integrin subunit (red) and DAPI (blue). **f** Quantification of red fluorescence intensity in control (white) and _B4galt1_⁻/⁻ MKs (green). $n = 3$ in each group. Data are expressed as mean ± SEM. Groups were compared using an unpaired Student's _t_-test, ****$p < 0.0001$. **g** Total lysates from control, _B4galt1_⁻/⁻, _Itgb1_^PF4+, and _B4galt1_⁻/⁻_Itgb1_^PF4+ platelets were subjected to SDS–PAGE and probed with anti-p-Tyr, pY397 FAK, pY576 FAK, total FAK, and β-actin mAbs. Densitometric quantification of immunoblots probed with **h** pY397 FAK ($n = 3$) and **i** pY576 FAK ($n = 6$) mAb, normalized for total FAK. Data are expressed as mean ± SEM. One-way ANOVA was used to compare each group, *$p < 0.05$; **$p < 0.01$; ***$p < 0.001$, ****$p < 0.0001$.

for Gal, ECL also binds to N-acetylgalactosamine (GalNAc) and fucosylated Gal (Fucα1–2 Gal)[46]. Hence, the residual ECL signal observed may be due to the additional lectin-binding specificities. By contrast, changes in terminal Gal and GlcNAc were low or insignificant in _B4galt1_⁻/⁻ CD45+ BM cells relative to control. LacNAc expression was not detectable in _B4galt1_⁻/⁻ MKs as judged by immunofluorescence microscopy of BM sections using ECL (Fig. 7f), thus corroborating the enzymatic activity

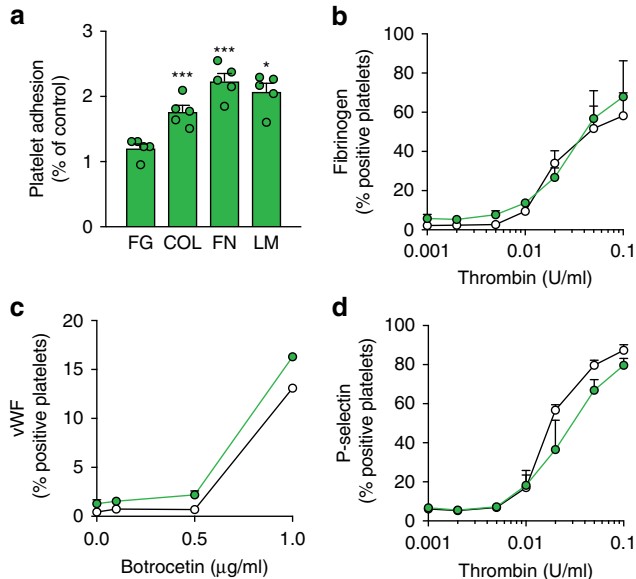

**Fig. 6** *B4galt1$^{-/-}$* **platelets adhere avidly to β1 integrin ligands in vitro.**
**a** Quantification of *B4galt1$^{-/-}$* platelet adhesion to fibronectin (FB), collagen (Coll), fibrinogen (FN), and laminin (LM)-coated coverslips. Values are normalized by substracting platelet adhesion to BSA-coated coverslips and expressed as fold over control platelets. $n = 5$ in each group. Data are expressed as mean ± SEM. Groups were compared using an unpaired Student's *t*-test, *$p < 0.05$. **$p < 0.01$, ***$p < 0.001$. **b–d** Flow cytometric quantification of percentage of control (black) and *B4galt1$^{-/-}$* (green) platelets bound to **b** oregon green-conjugated fibrinogen ($n = 3$), **c** FITC-conjugated anti-vWf antibody ($n = 2$), **d** FITC-conjugated anti-P-selectin antibody ($n = 3$) following in vitro platelet activation with **b**, **d** thrombin or botrocetin **c**. Data are expressed as mean ± SEM.

measurements and flow cytometry lectin binding. By contrast, other BM cells, presumably CD45$^+$ cells, expressed LacNAc. Thus, β4GalT1 expression is required for LacNAc synthesis in HSCs and MKs.

**TPO and CXCL12 regulate β4GalT1 and LacNAc synthesis in MKs.** MKs and HSCs share the dependence on common surface receptors, such as the TPO receptor MPL and the CXCL12 receptor CXCR4, lineage-specific transcription factors, and specialized signaling pathways. MPL controls HSC quiescence and MK differentiation, while CXCR4 regulates mobilization and migration of both cell types[11]. Further, CXCL12-mediated MKP migration to sinusoids depends on β1 integrin[11]. We reasoned that MPL and CXCR4 might regulate β4GalT1 activity in these cells and tested this hypothesis in fetal liver-derived MKs.

In the absence of TPO control fetal liver cell-derived MKs expressed only a few LacNAc-bearing polypeptides of ~130 and 270 kDa as determined by Western blotting using ECL (Fig. 7g). Stimulation of fetal liver cells with TPO upregulated the expression of LacNAc in MKs, as shown by the increased Gal exposure by multiple polypeptides with an apparent molecular weight of 25, 55 and 200–270 kDa. By contrast, the addition of CXCL12 alone had no detectable effect on LacNAc expression. Stimulation of fetal liver cells with TPO and CXCL12 increased LacNAc expression by polypeptides with a higher apparent molecular of 130 kDa and *B4galt1* mRNA levels (Fig. 7h) in control MKs. As expected, the polypeptide with a molecular weight of 130 kDa in *B4galt1$^{-/-}$* fetal liver cell-derived MKs had no detectable expression of LacNAc in the presence of TPO and CXCL12 (Fig. 7i).

## Discussion

β4GalT1 participates in the synthesis of Gal-containing oligosaccharides, including Lewis X (LeX) and its sialylated form, sialyl LeX (sLeX), the canonical binding determinant for selectins that serves as a critical ligand for lymphocyte trafficking[24,30,31]. Besides its role in LeX synthesis, other functions of β4GalT1 remain unknown. Here we describe a role for β4GalT1 as a specific regulator of β1 integrin posttranslational modifications in MKs and the role for DMS formation and thrombopoiesis (Fig. 8).

The β4GalT1-deficient mouse model used herein was generated by Lu et al.[30]. Our data show that about half of *B4galt1$^{-/-}$* mice in a mixed 129S1/C57BL/6J genetic background died before weaning right after birth, whereas mice bred in the C57BL/6J background had no surviving neonates, hence our data suggest that β4GalT1 plays a role during embryonic development. Our data show that mice bleed between E14.5 and E15.5 and exhibit profound thrombocytopenia. If bleeding contributes to the mortality of *B4galt1$^{-/-}$* mice remains to be determined. In agreement with published data, we observed epithelial cell proliferation of the skin and growth retardation of *B4galt1$^{-/-}$* mice[31].

An increase in TPO signaling, aging, and inflammation promotes HSC platelet bias[13–16]. The deletion of TGF-β and depletion of MKs enhances the loss of HSC quiescence and subsequent HSC proliferation[47,48]. Our data demonstrate that cytokines typically associated with HSC platelet bias, including TPO, PDGF, and IL-6, were comparable to controls or reduced (TGF-β) in the *B4galt1$^{-/-}$* BM environment. The high plasma TPO levels in *B4galt1$^{-/-}$* mice may promote platelet-biased HSC and MK numbers, thus providing a feasible explanation for normal TPO levels in *B4galt1$^{-/-}$* BMs.

Out of the seven β4GalTs, only β4GalT1 and β4GalT2 efficiently synthesized type-2 LacNAc[45]. Our data show that LSK cells and MKs expressed solely β4GalT1, while residual β4GalT activity was observed in WBCs, indicating that other blood lineages may compensate for β4GalT1 by expressing β4GalT2[31]. The data are consistent with the tissue-specific expression of β4GalTs, marking β4GalT1 responsible for the regulation of hematopoiesis. By contrast, β4GalT2 is the main β4GalT in the brain, corroborated by mouse models lacking β4GalT2 that develop motor-learning retardation and impaired motor coordination, but have no hematologic disorders[49]. Hence, our findings support the concept that HSCs and MKs share common cell regulatory pathways, such as MPL and CXCR4, including β4GalT1-dependent LacNAc synthesis and signals.

*B4GALT1* has been long considered a housekeeping gene. During thrombopoiesis, MPL and CXCR4 receptors activation upregulate *B4galt1*. The remarkable repertoire of hematopoietic cell-specific variations in transcript origination sites for the *B4GALT1* points to a more complex regulatory apparatus controlling gene transcription[27,28]. The 5′ end promoter region of *B4GALT1* contains multiple open chromatin sites and is rich in enhancer sequences with transcription factor-binding sites (TFBS), such as STAT2/3, TCF7L2, E2F1, and GATA1 that regulate thrombopoiesis and HSCs[27]. The wide variety of TFBS suggests that *B4GALT1* contains a super-enhancer region and thus emerges as a previously unrecognized gene that regulates cell identity, disease, and promotes multidrug resistance in hematologic cancers resistance[27,28]. Transcription factors involved in late MK differentiation and platelet formation, including GATA1, FOG1, FLI1, TAL1, RUNX1, NFE2, and E2F1[50], and the response to specific growth factors, such as TPO and CXCL12, play a pivotal role in the differentiation and maturation of MKs[11,51]. However, none of the transcription factor gene products increase during thrombopoiesis. By contrast, *B4galt1* mRNA expression and activity were highly increased in platelets compared to MKs, pointing to the

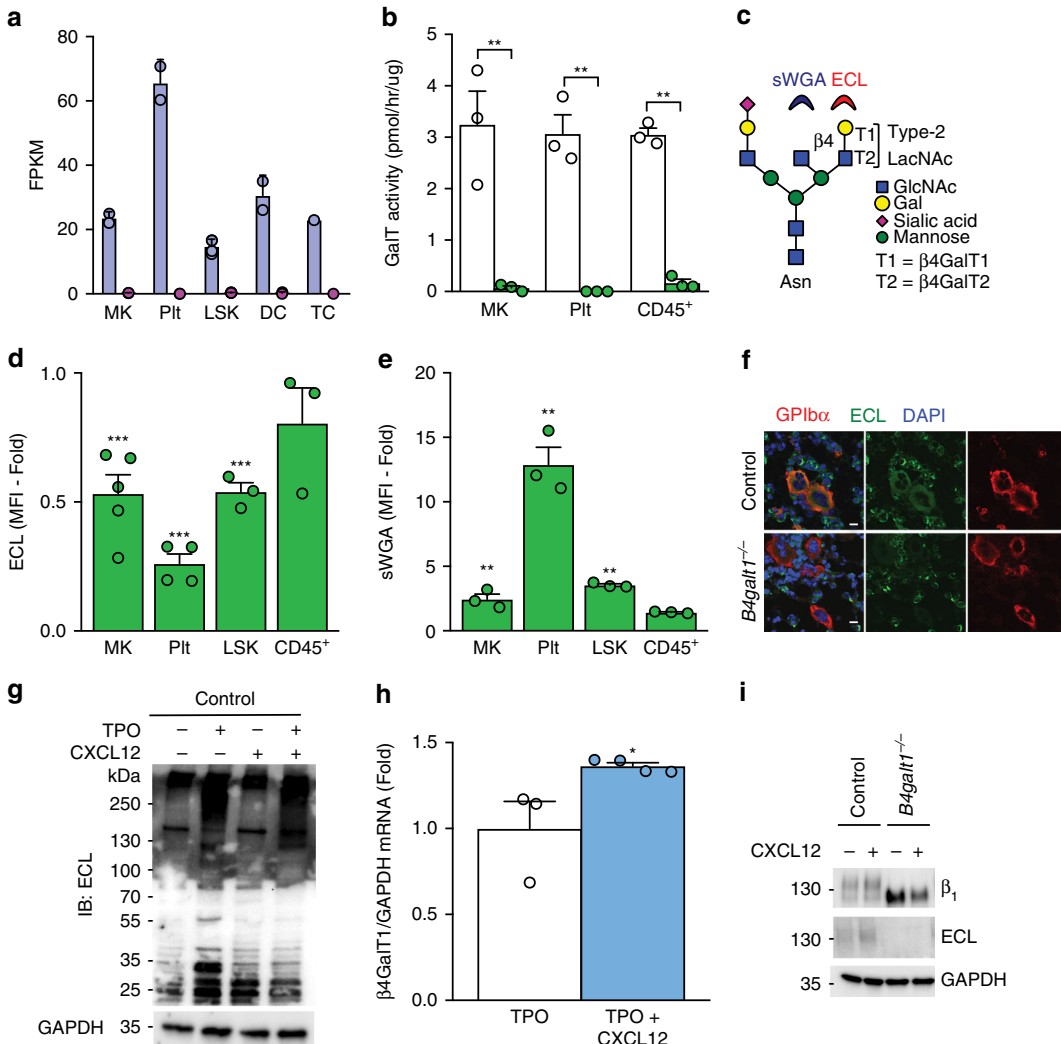

**Fig. 7 TPO and CXCL12 stimulate β4GalT1-dependent LacNAc synthesis in MKs. a** *B4galt1* (blue) and *B4galt2* (pink) Illumina mRNAseq expression levels in mouse MKs, platelets (Plt), LSK, dendritic (DC), and T (TC) cells (adjusted $p < 0.001$). **b** Galactosyltransferase activity towards the O-benzyl galactosyltransferase acceptor derivative GlcNAcα-O-Bn quantified in lysates of control (white) and *B4galt1*$^{-/-}$ (green) MKs, platelets (Plt), and BM CD45$^+$ cells. $n = 3$ in each group. Data are expressed as mean ± SEM. Groups were compared using an unpaired Student's *t*-test, $**p < 0.01$. **c** Scheme of a complex N-linked glycan bearing LacNAc structures constituted by GlcNAc and Gal which can be terminated by sialic acid. sWGA and ECL lectins bind to terminal GlcNAc and Gal, respectively. Terminal βGal **d** and βGlcNAc **e** moieties were measured by flow cytometry in blood platelets and BM MKs, LSK, and CD45$^+$ cells isolated from *B4galt1*$^{-/-}$ and control BMs using ECL and sWGA lectins, respectively. Data are expressed as mean fluorescence intensity (MFI) fold over control. $n \geq 3$ in each group. Data are expressed as mean ± SEM. Groups were compared using an unpaired Student's *t*-test, $**p < 0.01$, $***p < 0.001$. **f** Immunofluorescence of BM sections from control and *B4galt1*$^{-/-}$ mice were stained with anti-GPIbα mAb (red), ECL (green) and DAPI (blue). Scale bar: 10 μm. **g** Control and *B4galt1*$^{-/-}$ MKs were differentiated in vitro in the presence (+) or absence (−) of TPO, CXCL12, or both TPO and CXCL12, lysed and subjected to SDS–PAGE and probed with ECL and anti-GAPDH antibody as loading control. **h** *B4galt1* mRNA expression in control MKs in the presence (+) or absence (−) of CXCL12 $n \geq 3$ in each group. Data are expressed as mean ± SEM. Groups were compared using an unpaired Student's *t*-test, $*p < 0.05$. **i** Control and *B4galt1*$^{-/-}$ MKs were differentiated in vitro in the presence of TPO alone (−) and with addition of CXCL12 (+). Total MK lysates were subjected to SDS–PAGE and probed with anti-β1 integrin subunit antibody, ECL and anti-GAPDH antibody.

upregulation of the gene during late thrombopoiesis and platelet release, thus placing this gene as a regulator of thrombopoiesis at late stages, likely at the platelet release level.

Our data show that glycosylation via β4GalT1 regulates β1 integrin activity and platelet release: Lack of galactosylation in *B4galt1*$^{-/-}$ MKs renders β1 integrin hyperactive thus negatively affecting thrombopoiesis, and deletion of β1 integrin specifically in MKs restored thrombopoiesis. In HSCs and MKs, CXCL12 signaling activates β1 integrin to promote HSC migration and MK localization at sinusoids[11,52]. β1 integrin expression is indispensable for HSC homing and hematopoiesis during embryonic development and adult life[53]. By contrast, β1 integrin

expression is presumed dispensable in MKs and for thrombopoiesis, an assertion based on the fact that β1 integrin deletion does not affect thrombopoiesis and platelet count[3,4,54]. Gain-of-function mutations in the genes encoding for αIIb and β3 result in defective thrombopoiesis and thrombocytopenia[5–8]. The data strongly suggest that integrin activity needs to be tightly controlled in MKs to allow proper integrin–ECM interactions during the complex process of platelet release.

β1 integrin is regulated by differential glycosylation. For example, the presence of the N-glycan core structure is essential for integrin heterodimerization, stabilization of conformation, expression at the cell membrane, and interaction with ligands[41,55,56]. Branching of

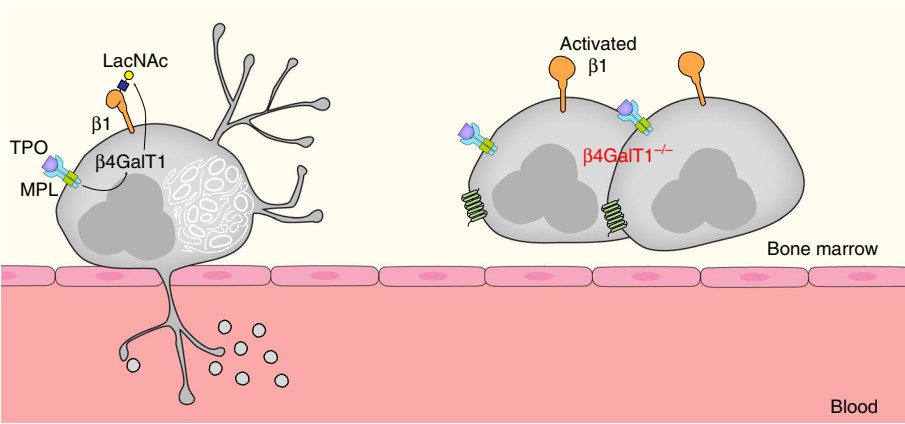

**Fig. 8 Scheme of how β4GalT1 regulates integrin function, DMS, and thrombopoiesis.** TPO stimulation regulates β4GalT1 expression and LacNAc synthesis of β1 integrin in MKs, indicated as multinucleated cells. In control MKs, β1 integrin function depends on LacNAc addition, thereby regulating the formation of the demarcation membrane system (DMS) and platelet release (thrombopoiesis). Impaired LacNAc synthesis in the absence of β4GalT1 leads to β1 integrin hyper-activity, defective DMS formation, and thrombopoiesis.

N-glycans $\alpha_3\beta_1$ or $\alpha_5\beta_1$ regulates cancer cell adhesion, migration, and metastasis[38,57–59]. Increased β4GalT1 expression and activity is significantly correlated with differentiation and mobilization of granulocytes and with the metastatic potential of specific cancer cells[37,60–64]. Modifications of LacNAc by sialic acid and Gal decorations on α4β1 or α5β1 negatively affect cell–cell and cell–matrix interactions, thereby increasing cancer cell mobility in colon adenocarcinoma, ovarian cancer, and hepatocellular carcinoma[37,64–69]. Control MKs express two β1 integrin glycoforms at 130 and 110 kDa, while platelets express the 130 kDa β1 glycoform primarily. By contrast, $B4galt1^{-/-}$ MKs and platelets only expressed the lower 110 kDa β1 integrin glycoform. Thus, β4GalT1 individually glycosylates the lower form of β1 integrin during thrombopoiesis. Lack of β4GalT1 affects the apparent molecular weight of α5, α6, and αIIb integrin subunits, suggesting functional impairment. Deletion of the β1 integrin subunit recovered thrombopoiesis, showing that the β1 integrin regulates the final stages of platelet formation and release. Differentially glycosylated integrin complexes by TPO and CXCL12, and ECM ligands may regulate platelet release in time and space. TPO and CXCL12 could sequentially promote β4GalT1-dependent LacNAc synthesis to fine-tune β1 integrin signaling and interaction with ECM glycoproteins to promote progenitor localization at the sinusoids and to regulate the formation of the DMS and platelet territories, a necessary prerequisite for platelet formation. The platelet-specific integrin αIIbβ3 appears to function normally in the absence of β4GalT1. This finding adds to the glycan specificity of the β1 integrin and their interaction with ECM components during the DMS-platelet forming process. The presence of ECM is necessary for optimized platelet release by human-induced pluripotent HSCs[70], a notion that supports the importance of ECM during platelet release. The notion that β1 integrin glycan–ECM interactions regulate this complex process provides mechanistic insight into thrombopoiesis. The continuous identification of glycosylation-related gene defects suggests an even more complex role for posttranslational modifications in thrombopoiesis. The here presented findings and published data support need for further investigations on the role of glycans for integrin function.

In summary, our findings establish a non-redundant role for β4GalT1 regulated by TPO and CXCL12 to govern MK β1 integrin glycosylation, integrin signaling, and thrombopoiesis, thereby affecting HSC homeostasis.

## Methods

**Mice.** $B4galt1^{-/-}$ mice[30] were provided by the Consortium for Functional Glycomics (www.functionalglycomics.org) and bred on a mixed background (129S1/

C57Bl/6). Wild type littermates were used as controls. $\beta_1^{fl/fl}$ transgenic mice, having loxP sites flanking exon 3 of the Itgb1 gene, PF4-Cre mice, expressing Cre recombinase under the control of the human PF4 promoter, UBI-GFP/BL6 mice[33], expressing a transgene coding for GFP under control of the human ubiquitin C promoter and NSG[34] immunocompromised mice were purchased from the Jackson Laboratory (Bar Harbor, ME, USA). $Itgb1^{fl/fl}$ mice were crossed with PF4-Cre mice in order to obtain mice lacking β1 integrin specifically in MKs and platelets. $B4galt1^{-/-}$ mice were crossed with $Itgb1^{fl/fl}$ PF4-Cre ($Itgb1^{PF4+}$) mice in order to obtain a double KO mouse lacking β4GalT1 and β1 integrin, the latter specifically in MKs and platelets. $B4galt1^{-/-}$ mice were crossed with UBI-GFP/BL6 mice to obtain $B4galt1^{-/-}$ UBI-GFP/BL6 mice to be used for BM transplant experiments.

**Study approval.** All experimental procedures involving animals were performed in compliance with all relevant ethical regulations applied to the use of small rodents, and with approval by the Animal Care and Use Committees (IACUC) at the Brigham and Women's Hospital and Harvard Medical School (Protocol Nos. 2017N000009 and 14-10-2748R) and at the Medical College of Wisconsin (Protocol No. AUA00005595).

**MK culture.** E14.5 fetal liver cells were cultured in DMEM medium supplemented with 10% fetal bovine serum, 1% of penicillin/streptomycin and cultured for 4 days in the presence of vehicle (PBS), recombinant TPO (50 ng/ml, R&D Systems, Minneapolis, USA), CXCL12 (100 ng/ml, R&D systems), or the combination of both. At day 4, mature MKs were enriched over a bovine serum albumin (BSA) density gradient and lysed for immunoblotting. When used for mRNA analysis, MKs were enriched over a BSA gradient, stained with biotin-conjugated CD41 antibody and further purified using anti-biotin magnetic microbeads (Miltenyi Biotech, San Diego, CA, USA). In order to assess pro-platelet production, MKs were enriched through a BSA density gradient on day 3. Enriched MKs were incubated for an additional 18 h to induce proplatelet formation and the release of nascent platelets. The percentage of MKs forming proplatelets was determined by counting proplatelet-forming and total MKs using an Olympus CKX41 microscope.

**Immunofluorescence of mouse bone sections.** Femurs were harvested and fixed into periodate-lysine-paraformaldehyde fixative (0.01 M sodium-M-periodate, 0.075 M L-lysine, 1% PFA) overnight at 4 °C. Femurs were rehydrated into 30% sucrose in phosphate buffer for 48 h, embedded in OCT (TissueTek®, Sakura Finetek USA, Torrance, CA, USA) and snap frozen in isopentane/dry ice mixture. Whole longitudinal single-cell-thick (5 μm) femoral cryosections were obtained using a Leica Cryostat and the Cryojane tape transfer system (Leica Microsystems, Wetzlar, Germany). Slides were thawed, rehydrated in PBS, permeabilized with T-TBS, blocked with 5% goat serum and incubated overnight with anti-GPIbα (1:100, catalog number M040, Emfret Analytics, Eibelstadt, Germany), anti-laminin (1:100, catalog number L9393, Sigma-Aldrich), and an anti-β1 integrin antibody that specifically binds to activated β1 integrins (1:100, catalog number 553715, clone 9EG7, Becton Dickinson, Franklin Lakes, NJ, USA). Slides were washed in PBS and incubated with alexa fluor-conjugated secondary antibodies (1:1000, Invitrogen). For evaluation of lectin binding non-permeabilized slides were blocked with Carbo-free blocking solution (Vector Laboratories) and incubated with FITC-conjugated ECL (1:3000, Vector laboratories, catalog number FL-1141). Immunofluorescence images were acquired by an Olympus FV10i confocal laser scanning microscope (Olympus, Deutschland GmbH, Hamburg, Germany). For quantification of MKs associated to sinusoids, the total number of MKs and number of MKs within 1 μm from a laminin+ vessel was counted in 10 different fields/sample.

Immunofluorescence data from control and *B4galt1⁻/⁻* BMs were analyzed for activated β1 integrin staining with Imaris Software (Bitplane, Switzerland). Surfaces were created around the β1 integrin staining in the MKs. Mean intensity of the β1 integrin fluorescence signal in each MK was exported for analysis in Excel.

**Mouse complete blood cell count.** Complete blood cell count was assessed by a Sysmex XT-2000i automated hematology analyzer (Sysmex, Kobe, Japan). Blood cell count in UBI-GFP donor transplanted mice and in E14.5 fetuses was assessed by flow cytometry as follows. Blood was incubated with anti-mouse PE-conjugated CD61 (1:100, catalog number 563523, Becton Dickinson), PE-conjugated TER-119 (1:100, catalog number 561033, Becton Dickinson), or PerCP-Cy5.5-conjugated CD45 (1:100, catalog number 561869, Becton Dickinson) for 20 min in the dark. 300 μl of PBS containing $10^4$ AccuCount-fluorescent beads (Spherotech) was added to each sample. Samples were analyzed in a FACSCalibur flow cytometer using CELLQuest software (Becton Dickinson). Platelets, red blood cells, and WBCs were identified based on the positivity for CD61, TER-119, and CD45, respectively, and their concentration was determined based upon the bead concentration. Donor-derived cells in transplanted mice were identified by their positivity for GFP. The gating strategy for each experiment sets is described in Supplementary Fig. 3.

**Platelet isolation.** Blood was obtained from anesthetized mice by retro orbital bleed in Aster–Jandl anticoagulant (85 mM sodium citrate, 69 mM citric acid, and 20 mg/ml glucose, pH 4.6). Platelet-rich plasma was obtained by centrifugation of the blood at $100 \times g$ for 8 min, followed by centrifugation of the supernatant and the buffy coat at $100 \times g$ for 6 min. After washing twice in washing buffer (140 mM NaCl, 5 mM KCl, 12 mM $Na_3C_6H_5O_7$, 10 mM glucose, and 12.5 mM sucrose, pH 6.0), platelets were resuspended in resuspension buffer (140 mM NaCl, 3 mM KCl, 0.5 mM $MgCl_2$, 5 mM $NaHCO_3$, 10 mM glucose, and 10 mM HEPES, pH 7.4) and counted by flow cytometry using 5.5-μm diameter SPHERO rainbow beads (Spherotech, Lake Forest, IL, USA).

**Platelet activation.** Washed platelets ($10^8$/ml) were activated with 0, 0.001, 0.002, 0.005, 0.01, 0.02, 0.05, 0.1 U/ml of human thrombin (Roche, Indianapolis, IN, USA) for 3 min at 37 °C. Platelets (5 μl) were then transferred into tubes containing 45 μl of a 1:100 dilution of FITC-conjugated anti-P-selectin antibody (1:100, catalog number 553744, Becton Dickinson) or a 1:300 dilution of oregon green FG (1:300, catalog number F7496, Invitrogen) and incubated for 20 min in the dark. Samples were finally diluted in PBS and analyzed in a FACSCalibur flow cytometer.

**vWf binding to platelets.** Washed platelets ($10^8$/ml) were incubated with 10% of autologous platelet poor plasma and 0, 0.1, 0.5, 1 μl of botrocetin (Sigma-Aldrich) for 4 min at 37 °C. Platelets (5 μl) were then transferred into tubes containing 45 μl of a 1:20 dilution of FITC-conjugated anti-vWF antibody (P150-1, Emfret Analytics) or a FITC-conjugated isotopic control. Samples were finally diluted in PBS and analyzed in a FACSDiva flow cytometer.

**Platelet adhesion.** Washed platelets ($10^8$/ml) were activated with 0.01 U/ml of human thrombin in the presence of 1 mM $Ca^{2+}$ and seeded onto glass coverslips coated with BSA (5%), FG (100 μg/ml), FN (50 μg/ml), LM (50 μg/ml), Type 1 Coll (50 μg/ml). After 30 min platelets were fixed with 4% paraformaldehyde and coverslips were washed three times in PBS. Adherent platelets were then stained with Alexa Fluor 568-conjugated phalloidin (Thermofisher). Coverslips were mounted with aqua-poly/mount (Polysciences, Warrington, PA, USA). Immunofluorescence images were acquired by an Olympus FV10i confocal laser scanning microscope (Olympus, Deutschland GmbH, Hamburg, Germany). The number of platelets/field was counted and numbers were adjusted by subtracting aspecific values counted in BSA-coated slides.

**Immunoblotting.** Mouse platelet and MK protein lysates were subjected to gel electrophoresis (SDS–PAGE) and transferred to polyvinylidene fluoride (PVDF) membrane (BioRad, Hercules, CA, USA). Membranes were probed with: biotinylated ECL (1:3000, catalog number B-1145, Vector Laboratories); anti-mouse integrin β1 polyclonal antibody (1:5000, catalog number ab183666, Abcam); anti-αIIb (1:5000, clone 386627, catalog number MAB7616, R&D systems); anti-α5 (1:1000, polyclonal, catalog number 4705S, Cell Signaling Technology, Massachusetts, USA); anti-α6 (1:1000, polyclonal, catalog number 3750, Cell Signaling Technology); anti-mouse β3 integrin (1:5000, polyclonal, catalog number 4702, Cell signaling Technology); anti-GPIbα (1:5000, clone Xia.G7, catalog number M042-0, Emfret analytics); anti-mouse FAK (1:1000, clone D2R2E, catalog number 13009, Cell Signaling Technology); and anti-phospho-FAK (Tyr397 clone D20B1, catalog number 8556, and Tyr576/577, polyclonal, catalog number 3281, 1:1000, Cell Signaling Technology); anti-phosphotyrosine (1:4000, clone 4G10, catalog number 05-321 Millipore, USA), anti-GAPDH (1:5000, clone 14C10, catalog number 2118, Cell signaling Technology). Immunoreactive bands were detected by horseradish peroxidase-labeled streptavidin (1:3000, Cell Signaling Technology) or horseradish peroxidase-labeled secondary antibodies (BioRad), using enhanced chemiluminescence reagent (Millipore). Pre-stained protein ladders (BioRad) were used to estimate the molecular weights. Band intensities from individual western

blots were quantified by densitometric analysis using software ImageJ. The uncropped blots can be found in the Source Data file and in Supplementary Figs. 4 and 5.

**Immunoprecipitation.** Platelet lysates were incubated overnight at 4 °C with 30 μg/ml of anti-mouse integrin β1 antibody. Platelet lysates were then incubated with sepharose beads (GE Healthcare) for 2 h at 4 °C. Sepharose beads were washed three times in lysis buffer and resuspended in lamely buffer. Samples were then subjected to SDS–PAGE, transferred to PVDF membrane and probed for β1 integrin and ECL.

**Quantitative real time PCR.** In vitro differentiated mouse MKs were purified using a immunomagnetic bead technique. Total RNA isolated using the TRIZOL Reagent (Invitrogen, Carlsbad, CA). Retrotranscription (RT) was performed using the iScript™ cDNA Synthesis Kit according to the manufacturer instructions (BioRad). For quantitative Real Time PCR, cDNA was amplified in the presence of primers listed in supplementary Table 2. GAPDH was amplified as internal control.

**In vivo mouse platelet survival.** In vivo platelet survival was performed by retro-orbitally injecting Biotin-NHS (Calbiochem) into WT and *B4galt1⁻/⁻* mice (24 mg/kg body weight). Mice were bled 24, 48, and 72 h after injection. Platelets were stained with PE-conjugated streptavidin (Becton Dickinson) and the percentage of biotin-positive platelets was quantified by flow cytometry.

**Ploidy of mouse BM MKs.** MK ploidy was evaluated by flow cytometry. BM cells were flushed with washing buffer (HBSS, 0.38 sodium citrate, 1 mM adenosine, 1 mM theophylline, 5% FBS) and red blood cells were lysed through addition of ACK lysing buffer (0.15 M $NH_4Cl$, 10 mM $KHCO_3$, 0.1 mM $Na_2EDTA$, pH 7.4). BM cells were fixed in 70% ethanol, washed twice and resuspended in washing buffer. Cells were incubated for 30 min on ice with FITC-conjugated anti-CD41 (1:100, catalog number 553848, Becton Dickinson) or rat-IgG1 κ isotype control (Becton Dickinson) antibody. After washing twice, cells were treated with 0.5 mg/ml ribonuclease A (Sigma-Aldrich), followed by DNA staining with 0.05 mg/ml propidium iodide (PI). Cells were analyzed by flow cytometry. To assess MK membrane glycoprotein expression, BM cells were stained with PE-labeled CD41 antibody and FITC-conjugated anti-β3 integrin (CD61, 1:100, catalog number 563523, Becton Dickinson), anti-β1 integrin (1:100, catalog number 555005, Becton Dickinson), anti-GPIbα (1:100, M042-1, Emfret Analytics) or isotypic control and incubated for 30 min in the dark.

**HSPC analysis.** HSPC analysis of control, *B4galt1⁻/⁻* and of BM-transplanted mice was performed by flow cytometry as previously described. BM and spleen cells were collected and prepared for staining by erythrocyte lysis (BD Pharm Lyse; Becton Dickinson) and homogenization through a 70-μm filter. Cells were stained in ice-cold PBS-containing 2% FBS using the following antibodies: lineage cocktail containing 1:100 dilution of CD3e (catalog number 48-0031-82, Ebioscience), CD5 (catalog number 48-0051-82, Ebioscience), Ter-119 (catalog number 48-5921-82, Ebioscience), Gr-1 (catalog number 48-5931-82, Ebioscience), Mac-1 (catalog number 48-0112-82, Ebioscience), and B220 (catalog number 558108, BD Pharmingen), c-Kit (clone 2B8, catalog number 17-1171-82, eBioscience), Sca-1 (Clone D7, catalog number 25-5981-82,eBioscience), CD150 (Clone TC15-12F12.2, catalog number 115912, Biolegend, San Diego, CA, USA), CD48 (clone HM48-1, catalog number 103432, Biolegend), CD34 (clone RAM34, catalog number 11-0341-85, eBioscience), CD16/32 (clone 93, catalog number 12-0161-83, Ebioscience), CD41 (clone MWReg30, catalog number 17-0411-82, Ebioscience), and CD105 (clone MJ7/18, catalog number 25-0151-82, Ebioscience). DAPI (Invitrogen) was used for dead cell discrimination. In a subset of experiments BM cells were stained also for CD41, CD29 (clone Ha2/5, Becton Dickinson) or with ECL and sWGA lectins (Vector Laboratories). For analysis of HSC cycling samples were fixed and permeabilized with cytofix/cytoperm solutions (Becton Dickinson) followed by staining with anti-Ki67 (652404 Biolegend) and Hoechst 33342. The gating strategy for each experiment sets is described in Supplementary Fig. 3b, c.

Samples were analyzed using an LSR II (Becton Dickinson). Post-acquisition analysis of data was performed with FlowJo software V9.2.3.

**Colony-forming cell assay.** For colony-formation assays, 20,000 BM mononuclear cells were plated in duplicate in a methyl cellulose-based medium containing recombinant cytokines (MethoCult #03434, Stem Cell Technologies) and incubated at 37 °C, in 5% $CO_2$. Colony number was counted following 10–12 days in culture.

**BM transplant.** Due to the mixed background of *B4galt1⁻/⁻* mice, WT, and *B4galt1⁻/⁻* UBI-GFP BM cells were injected into sub-lethally irradiated (250 cGy) NSG recipients ($2 \times 10^6$/mouse)[34]. Blood was drawn from recipient mice 1, 2, 4, 8, 12 and 16 weeks post-transplantation to evaluate peripheral blood cell count as described above. Transplanted mice were then sacrificed for BM analysis.

C57BL6/J E14.5 UBI-GFP fetal liver cells were injected into lethally irradiated (1000 cGy) WT recipients ($5 \times 10^6$/mouse). Fetal liver cells isolated from recipient mice were used as supportive cells ($0.5 \times 10^6$/mouse).

**Galactosyltransferase assay**. An enzymatic assay was used to evaluate galacto-syltransferase activity in cell lysates[71]. Transfer of a radiolabeled sugar nucleotide donor, UDP-[$^3$H]Gal (American Radiolabeled Chemicals, MO, USA), to an O-benzyl galactosyltransferase acceptor derivative structure, GlcNAcα-O-Bn (Toronto Research Chemicals, Canada) was measured. The incubation mixture (3 μl) contained 25 mM Tris–HCl pH 7.5, 1 mM each of $MnCl_2$ and $MgCl_2$, 2.5 mM acceptor, 1.25 μM radiolabeled sugar-nucleotide donor, 5 μg/μl BSA and 0.5% (W/V) Triton X-100. Incubation was carried out at 37 °C for 1.5 h. Transfer of radioactively labeled sugars to benzyl-derivatized acceptors was monitored by scintillation counting after removal of unreacted radiolabeled sugar-nucleotides and by-products by C18-reverse phase chromatography (SepPak).

**Transmission electron microscopy**. BM cells were fixed in 2.5% glutaraldehyde in 0.1 M sodium cacodylate buffer, pH 7.4, for 1 h at room temperature, washed, post fixed in 1% osmium tetroxide ($OsO_4$) on ice for 1 h, washed, moved through graded alcohols and embedded in EMBed 812 epoxy resin. 70-nm sections were cut onto a RMC 'Powertome' stained with uranyl acetate and lead citrate then viewed in a Hitachi H600 TEM. Images were obtained using AMT camera system.

**Cytokine array**. BM supernatants were generated by crushing 2 tibias and 1 femur in the presence of 400 μl of PBS in a sterile mortar. Samples were centrifuged (600 × g for 10 min), protein levels were quantified using Pierce™ BCA Protein Assay Kit and 100 μg of total protein were used. Cytokine levels were determined using Proteome Profiler Mouse XL Cytokine Array (R&D systems) according to the manufacturer's instruction. Quantification was done by ImageQuant(TM) TL (GE Healthcare, Chicago, IL, USA) and the cytokine levels were expressed as arbitrary units.

**Illumina RNAseq analysis**. To compare the expression level of *B4galt1* and *B4galt2* across murine blood cell types we analyzed publically available Illumina RNAseq datasets. All libraries were normalized using DESeq2 v1.18.1 and FPKM values calculated from these normalized libraries were used in Fig. 5e. For additional information and references, see Supplementary Methods and References.

**Statistical analysis**. All experiments were performed at least in triplicate and data are represented as mean ± standard error of the mean (SEM). Numeric data were analyzed using one-way ANOVA analysis of variance followed by Bonferroni adjustment for multiple comparisons. Two groups were compared by the two-tailed Student's unpaired *t*-test. The significance of data was assessed using the GraphPad Prism 5 software. Differences were considered as significant when $p < 0.05$. Different levels of significance are indicated as $*p < 0.05$, $**p < 0.01$, $***p < 0.001$.

**Reporting summary**. Further information on research design is available in the Nature Research Reporting Summary linked to this article.

## Data availability

The data generated and/or analyzed during the current study are available from the corresponding author on reasonable request. The source data underlying Figs. 1–7 and Supplementary Figs. 1 and 2 are provided as a Source Data file.

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

## Acknowledgements

This work was supported by National Institutes of Health grants R01 HL134829 (S.G.), R01 HL089224 (K.M.H.), P01 HL107146 (K.M.H.), K12 HL141954 (K.M.H.), R01 HL126743 (H.F.), R01 AI140736 (J.T.L.), American Society of Hematology bridge grants (S.G., H.F.). We thank Max Adelman, Ann Mullally and Antonija Jurak Begonja for helpful discussions and help with experimental design.

## Author contributions

S.G. designed and performed research, collected, analyzed and interpreted results, and assisted with writing the manuscript. M.M.L.-S., L.R. and C.A.D.B. designed and performed research. R.B. analyzed and interpreted results. H.F., J.T.L. and A.B. interpreted results and assisted with manuscript writing. K.M.H. designed research, analyzed and interpreted results, and wrote the manuscript.

## Competing interests

The authors declare no competing interests.
