## [Peer Review File · Nature Communications]

Reviewers' comments:

Reviewer #1 (Remarks to the Author):

General comments

This is an interesting manuscript that defines a new mechanism involved in control of platelet homeostasis. Although the patient cohort analyzed in this study is too small to draw conclusions, the findings could still have clinical implications for patients with primary myelofibrosis. Some issues should be addressed, outlined in my specific comments below.

Specific comments

1. The authors suggest that $\beta 4\text{GalT1}^{-/-}$ mice die due to hemorrhage. However, these mice also have $\beta 1$ integrin abnormalities, which might contribute to lethality. No data are provided to substantiate bleeding as a cause of fetal lethality in $\beta 4\text{GalT1}^{-/-}$ mice.

2. The authors suggest that activation of $\beta 1$ integrin is responsible for reduced TGF- β secretion. However, they do not show, whether or not loss of $\beta 1$ integrin in MKs of $\beta 4\text{GalT1}^{-/-}$ mice corrects TGF- β release. This appears essential to support the proposed hypothesis.

3. The MK/platelet phenotypes of $\beta 4\text{GalT1}^{-/-}$ mice may not be due to loss of $\beta 4\text{GalT1}$ in the platelet/MK lineage. The design of the transplant experiments does not rule out the contribution of hematopoietic cells outside the MK lineage.

4. The hypothesis that $\beta 4\text{GalT1}$ -dependent glycosylation aggravates myelofibrosis is interesting, yet the patient cohort analyzed in this study is too small to draw conclusions. The authors could test this hypothesis in a mouse model of primary myelofibrosis.

5. The function of platelets produced in $\beta 4\text{GalT1}^{-/-}$ mice should be defined.

Reviewer #2 (Remarks to the Author):

The authors presented a novel role of $\beta 4\text{GalT1}$ in thrombopoiesis by regulating $\beta 1$ -integrin activity through its glycosylation. The impaired thrombopoiesis could induce reduced TGF- β release in the BM environment, resulting in platelet-biased HSC. A variety of experiments were well performed and resulting data support their model (Figure 8). However, several concerns remained to be addressed to build up their model.

Major points

- 1) In the BM of $\beta 4\text{GalT1}^{-/-}$ mice, the number of MKs increased 3.7-fold, whereas they showed severe thrombocytopenia, 1/5-fold reduction of platelets in PB in Table 2. Although the authors showed several data in Figure 4 that supported the impaired thrombopoiesis in $\beta 4\text{GalT1}^{-/-}$ MKs, further data and more explanation are needed to explain the big difference between MKs in the BM and platelets in PB.
- 2) In Figure 3F, $\beta 4\text{GalT1}^{-/-}$ BM cells could populate myeloid-lineage cells and MKs, but not B and T cells in the recipient BM after non-competitive transplants. However, $\beta 4\text{GalT1}^{-/-}$ BM cells showed normal ratio of B and T cells in Figure 2A, although T cells were slightly decreased. Please explain this discrepancy. Homing and engraftment of HSCs might be disturbed.
- 3) In non-competitive transplants, ratio of GFP+ $\beta 4\text{GalT1}^{-/-}$ MKs at sinusoid was about half in Figure 3J, while ratio of GFP+ $\beta 4\text{GalT1}^{-/-}$ MKs in the whole BM increased 4-fold in Figure 3G, suggesting that the number of GFP+ $\beta 4\text{GalT1}^{-/-}$ MKs at sinusoid was 2-fold higher. However, GFP+ $\beta 4\text{GalT1}^{-/-}$ platelets in PB was less than 10%. Please explain why GFP+ $\beta 4\text{GalT1}^{-/-}$ platelets profoundly decreased, as asked in 1).
- 4) $\beta 1$ -integrin was activated in $\beta 4\text{GalT1}^{-/-}$ MKs in Figure 5A-E. However, there was no direct evidence that active $\beta 1$ -integrin was agalactosylated. Please compare the activity of fully galactosylated and agalactosylated $\beta 1$ -integrin.
- 5) In Figure 6B, GalT activity was not detected in $\beta 4\text{GalT1}^{-/-}$ MKs and platelets at all. However, MFIs of ECL signals were about half and 1/4 in $\beta 4\text{GalT1}^{-/-}$ MKs and platelets, respectively in Figure 6D. Please explain this discrepancy.
- 6) ECL signals markedly increased in MKs by TPO and CXCL12 in Figure 6G, while $\beta 4\text{GalT1}$ mRNA levels slightly increased (1.3-fold) by TPO and CXCL12 in Figure 6H. Can the slight increase of $\beta 4\text{GalT1}$ mRNA account for the markedly increase of ECL signals?
- 7) ECL signals were examined on the surface and in whole cell lysates of MKs in Figure 6, but not on $\beta 1$ -integrin. ECL signals on $\beta 1$ -integrin should be examined.

Minor points

- 1) Global examinations about $\beta 4\text{GalT1}^{-/-}$ mice, such as birth rate, weaning rate and body weight were also described in a previous paper (EMBO J 16: 1850-57, 1997). Please refer to this paper and discuss.
- 2) Please show whole gel images of Figure 3A, 5A and 5B as supplemental figures.
- 3) Since PF4 signals differed between control and $\beta 4\text{GalT1}^{-/-}$ MK releasate, TGF- β /PF4 values are not appropriate in Figure 3D.
- 4) In Figure 6G, it appears that control gel was strongly exposed, while $\beta 4\text{GalT1}^{-/-}$ gel was weakly exposed, because background signals were different. Both samples should be loaded in the same gel and exposed at the same level.
- 5) In the text, Figure S2 is missing and Figure S3, which is not attached, is described.
- 6) Figure S2 and its legend are incompatible.
- 7) In Experimental procedures, immunoblotting, before ECL lectin blotting, the membrane must be treated with sialidase to remove sialic acids. Without sialidase treatment, ECL cannot bind to galactosyl residues.
- 8) On page 9, line 22, "two additional polypeptides" should be "three additional polypeptides".
- 9) On page 13, line 2, "increased control" might be "increased more than control".
- 10) On page 27, title of Figure 2 legend, " $\beta 4\text{GalT1}^{-/-}$ " should be " $\beta 4\text{GalT1}$ ".
- 11) On page 28, title of Figure 4 legend, " $\beta 4\text{GALT1}$ " should be " $\beta 4\text{GalT1}$ ".
- 12) In Figure 8, "Platelet-biased HSC" might be in the right $\beta 4\text{GalT1}^{-/-}$ model.

Reviewer #3 (Remarks to the Author):

In this study the authors have studied the role of beta4GALT1 on hematopoiesis using beta4GALT1 knock out mice. The authors have focused their study on the HSC compartment and megakaryopoiesis as these mice have a profound thrombocytopenia and that the role of beta4GALT on neutrophils and the erythroid lineage have been previously studied.

General comment

There is a strong central message which is beta4GALT1 induces a defect in terminal megakaryocyte differentiation associated with a poor development of DMS leading to a profound defect in

proplatelet formation. This defect is related to an activation of beta 1 integrin as a consequence of a defect in glycosylation.

The other parts of the papers are not so well demonstrated and in some way are weakening the central message. Thus in my opinion the paper must be reorganized and the mechanism by which beta 1 activation impairs megakaryocyte differentiation must be further studied while the other parts could be omitted.

Specific comment

- A significant part of the paper is dedicated to the hematopoietic stem cell compartment and is only based on the immunophenotype of the HSC compartment. In fact to characterize this compartment and the presence of a megakaryocyte/platelet bias it will be necessary to perform ScRNA-seq and transplantations. However transplantation in this mixed genetic background is difficult and the transplantations in NGS mice are not the best manner to perform a detailed analysis. Furthermore as half of the homozygous knock out mice died, we do not know if a selection occurs which could be hematopoietic. Therefore such studies will require to have a new model of mice i.e. an inducible knockout. In the text it is not described what are the control mice in this mixed genetic background? Is it wild type littermate? In addition as the expression of CD41 on HSC greatly increases with aging, at what age was performed the studies and had the knock out and control mice the same age? Why the authors did not study the spleen as in the initial description of the mice an extramedullary hematopoiesis was present?

- The part on the TGF beta is partially convincing. Why have the authors studied the TGF beta level by western blots in megakaryocyte lysates and in the megakaryocyte supernatants? I think that an Elisa would have been more quantitative and could have also permitted to discriminate between the active and inactive form. It is unclear why there is an active form already in the MK lysate. It is usually considered that TGF beta 1 is stored under an inactive form and activated after secretion. Most of the studies have been performed on in vitro cultured fetal megakaryocytes, which could be an important bias in comparison to the bone marrow. Why the level of TGF beta has not been studied in the plasma and the spleen? The manner to quantify TGF beta in the culture supernatant is also difficult to understand. This is done in comparison to PF4. However in the text the authors assess that the level of secreted PF4 is increased for beta4GALT1 minus megakaryocytes. Finally the ratio between the active and inactive form in beta4GALT1 mice is not modified, thus there is no defect in TGF beta1 activation, but may be a defect in synthesis. Did the authors study the level of TGF beta1 transcript in megakaryocytes?

- The third part on beta beta4GalT1 and megakaryocyte differentiation is well performed and convincing for the role of beta1 integrin activation. However the precise mechanism remains to be understood. How beta 1 integrin activation impairs the development of DMS? Are some Rho GTPases implicated in this inhibition of megakaryopoiesis? It will be interesting to study in more details how these knock out megakaryocytes adhere to different matrix such as collagen or fibronectin and what are the changes in the cytoskeleton as well as in cell migration. How do the authors explain that platelets have a normal half life? Is the level of TPO normal in the plasma not only in bone marrow supernatant?

- The fourth part on the regulation by TPO and CXCL12 is also not convincing for several reasons. In Figure 6 H there is not the level of beta4GALT1 in the absence of TPO or in the presence of CXCL12 alone. In addition AZD 1480 is used at high concentration and at this concentration it does not only inhibit JAK1/JAK2 but also Aurora Kinases. Thus lower doses and the use of other inhibitors are required. Did the authors stimulate megakaryocyte by IL6 to induce the JAK/STAT pathway? As CXCL12 has no effect alone, the absence of effect of CXCL12 with TPO in the presence of a JAK inhibitor does not demonstrate that CXCL12 requires JAK2. Did the authors study the level of beta4GALT1 in c-mpl constitutive knock out megakaryocytes, which will be a manner to know if beta4GALT1 is regulated by TPO? If their hypothesis is true, how the authors explain that the deletion of c-mpl or Jak2 specifically in megakaryocytes with the PF4-Cre induce an important thrombocytosis? If they were right it is expected a profound thrombocytopenia.

- The work on patients with PMF is a totally different topic. Indeed in this part the authors show that in these patients with a constitutive JAK2 activation the Beta4GALT1 is increased and in some way correlates with the level of fibrosis. So the authors have first to show that an increase in Beta4 GALT1 level using transgenic mice or lentiviral vector will induce a myelofibrosis or at least an increase of active TGF beta1 release before to study patients. In addition 6 of the patients described in Table 3 have not a PMF because they have either no fibrosis (1 patient) and 5 a grade 1. Using the WHO classification, at least a grade 2 fibrosis is required for the diagnosis of PMF. I think that this part is not at all relevant for this paper.

- Many references are review, it will be much better to add the specific references.

Reviewer #4 (Remarks to the Author):

Overall, the manuscript is well-organized and well-written.

This reviewer has the following questions to be answered.

1. What is the role(s) of integrin $\beta 1$ on the CXCL12 (SDF1), FGF1, and TGF β cytokine signaling?
2. There is no mechanism of why $\beta 4$ Galt1 knockout mature staged MKs are NOT located at sinusoid site. How $\beta 1$ integrin with LacNAc (glycosylated structure) regulated the location of MKs?

3. What is the counter-subunit α of $\beta 1$ integrin? $\alpha 2\beta 1$ for collagen I, $\alpha 4\beta 1$ for VCAM-1/Fibronectin, $\alpha 6\beta 1$ for laminin? If all integrins are involved in this story, the data should provide the cases of individual ECM and specified integrin cascade for this story.
4. Why $\beta 1$ -integrin subunit deletion significantly improved circulating platelet count to 70%?
5. It is pretty difficult to understand the following statement, "Together, the data shows that absence of $\beta 4\text{GalT}1$ does not affect HSC quiescence but promotes HSC numbers and platelet-bias due to impaired thrombopoiesis and secretion of cytokines from $\beta 4\text{GalT}1^{-/-}$ MKs." Because in MK-HSC niche system diminished TGF β levels might induce the escape from dormancy of HSC leading to cell cycle acceleration at HSC/MPP levels. Why did you conclude that absence of $\beta 4\text{GalT}1$ does not affect HSC quiescence? There are lacks of information regarding interaction between several cytokine influences in this manuscript. This is weak point.
6. Another weak point is that the author did not use conditional $\beta 4\text{GalT}1$ knockout mice specifically targeting megakaryocyte to directly evaluate the effect of $\beta 4\text{GalT}1$ knockout on platelet release from MKs in adult mice.
7. The glycosylation of $\beta 1$ -integrin by $\beta 4\text{GalT}1$ is required for platelet release through suppression of integrin active form, but not megakaryocyte maturation? Or, stable $\beta 4\text{GalT}1$ structure somehow influence the inhibition of active form of $\beta 1$ integrin?
8. In this case, inside-out signaling or direct outside-in signaling of integrin is involved?
9. This reviewer is also curious in general that glycosyltransferases is the most critical switch of platelet shedding initiation from MKs?
10. It is assumed that glycosylation takes place for protein biosynthesis at Golgi apparatus. How glycosylation is completed in $\beta 1$ integrin structure at Golgi?
11. There are any differences of PLT functionalities between $\beta 4\text{GalT}1$ KO and WT PLTs?

ANSWER TO REVIEWERS

REVIEWER #1

1. The authors suggest that $\beta 4\text{GalT1}^{-/-}$ mice die due to hemorrhage. However, these mice also have $\beta 1$ integrin abnormalities, which might contribute to lethality. No data are provided to substantiate bleeding as a cause of fetal lethality in $\beta 4\text{GalT1}^{-/-}$ mice.

According to reviewer's suggestion, the sentence about mouse lethality has been rephrased as follows: "Evidence supports hemorrhage at embryonic stage E14.5-E15.5 in $\beta 4\text{GalT1}^{-/-}$ fetuses, however if bleeding is a cause of $\beta 4\text{GalT1}^{-/-}$ mouse mortality at this stage remains to be determined." (Page 5, line 11). Additionally, we provided more evidence for bleeding in Figure 1.

2. The authors suggest that activation of $\beta 1$ integrin is responsible for reduced TGF- β secretion. However, they do not show, whether or not loss of $\beta 1$ integrin in MKs of $\beta 4\text{GalT1}^{-/-}$ mice corrects TGF- β release. This appears essential to support the proposed hypothesis.

As suggested by multiple reviewers, the data pertaining to primary myelofibrosis (PMF) and TGF- β released by megakaryocytes dilute the message on the role of $\beta 1$ integrin and thrombopoiesis. Hence the data have been removed from this manuscript. The focus of the manuscript is now on $\beta 1$ integrin and $\beta 4\text{GalT1}$ dependent glycosylation as a regulator of thrombopoiesis.

3. The MK/platelet phenotypes of $\beta 4\text{GalT1}^{-/-}$ mice may not be due to loss of $\beta 4\text{GalT1}$ in the platelet/MK lineage. The design of the transplant experiments does not rule out the contribution of hematopoietic cells outside the MK lineage.

We apologize that the transplants were not explained more extensively. Bone marrow transplants were performed to exclude the effect of non-hematopoietic components affecting thrombopoiesis by $\beta 4\text{GalT1}^{-/-}$ megakaryocytes. We performed two different types of transplants: 1) using the mixed background (C56BL/6/129) bone marrow HSCs, which were transplanted into semi-lethally irradiated mice NSG mice, and 2) we transplanted fetal liver HSCs isolated from C57BL/6 embryos at E14.5 which were transplanted into C57BL/6 lethally irradiated mice in the presence of supporting cells. To clarify if the thrombopoiesis defect in $\beta 4\text{GalT1}^{-/-}$ megakaryocytes is intrinsic, we now added *total* platelet numbers to the figure, which demonstrate normal platelet count produced by control megakaryocytes in both transplant model. Thus, in both models, control platelets were produced normally suggesting that bone marrow cells and components are sufficient to support thrombopoiesis. The data are now shown in Fig. 3, p.7, line 1-8. *In vitro* data further support this notion, as proplatelet formation by cultured MKs was severely impaired.

Furthermore, we fully agree with the Reviewer that bone marrow transplants do not completely rule out the effect of other hematopoietic cells in the bone marrow of $\beta 4\text{GalT1}^{-/-}$ mice. In order to evaluate this, a conditional $\beta 4\text{GalT1}^{-/-}$ mouse, lacking $\beta 4\text{GalT1}$ specifically in the megakaryocyte lineage, is needed. The generation of the conditional $\beta 4\text{GalT1}^{-/-}$ mouse requires a considerable amount of time and unfortunately is not available at this time. To partially circumvent this problem, we generated $\beta 4\text{GalT1}^{-/-}$ mice that specifically lack $\beta 1$ integrin in MKs, a maneuver that normalized platelet count. Thus, these data support the notion that the defect in thrombopoiesis in $\beta 4\text{GalT1}^{-/-}$ mice is intrinsic to MKs.

4. The hypothesis that $\beta 4\text{GalT1}$ -dependent glycosylation aggravates myelofibrosis is interesting, yet the patient cohort analyzed in this study is too small to draw conclusions. The authors could test this hypothesis in a mouse model of primary myelofibrosis.

We agree with the majority of the Reviewers that we need to increase the number of primary myelofibrosis (PMF) patients enrolled in the study. As this is not possible in a timely manner, we removed the data that pertain to PMF.

We generated two state-of-the-art mouse models of PMF obtained 1) by transplanting HSCs over-expressing thrombopoietin and 2) by chronic administration of the thrombopoietin receptor agonist Romiplostim. Despite a clear increase in terminal galactosylation by several cell types and extracellular matrix components following the development of myelofibrosis, both mouse models did not phenocopy the megakaryocyte phenotype found

in patients with PMF, hence hampering our investigation. The data require more investigation and will be included in a future manuscript.

5. The function of platelets produced in $\beta 4\text{GalT1}^{-/-}$ mice should be defined.

$\beta 4\text{GalT1}^{-/-}$ platelet function data have been added in Figure 6. We have evaluated α -granule release (P-selectin surface exposure) and activation of the fibrinogen receptor $\alpha\text{IIb}\beta 3$ following thrombin activation *in vitro* with no measurable differences between the phenotypes. Von Willebrand Factor receptor (GPIb α / β /IX/V) function was evaluated by ristocetin-induced von Willebrand factor (vWf) binding to platelets and showed no apparent difference in vWf binding between the phenotypes.

To specifically investigate $\beta 1$ integrin function, we performed platelet adhesion studies using $\beta 1$ -integrin ligands (fibronectin, collagen, and laminin) *in vitro* (see also reviewer 4). The data clearly show that $\beta 4\text{GalT1}^{-/-}$ platelets adhere more avidly to all of the tested $\beta 1$ integrin ligands, and bind normally to the $\beta 3$ integrin ligand, fibrinogen, further supporting notion that $\beta 1$ integrin is hyperactive in the absence of $\beta 4\text{GalT1}$ mediated glycan synthesis. Additionally, we have now provided evidence that $\beta 1$ integrin is agalactosylated in the absence of $\beta 4\text{GalT1}$ (Figure 6).

REVIEWER #2

1. In the BM of $\beta 4\text{GalT1}^{-/-}$ mice, the number of MKs increased 3.7-fold, whereas they showed severe thrombocytopenia, 1/5-fold reduction of platelets in PB in Table 2. Although the authors showed several data in Figure 4 that supported the impaired thrombopoiesis in $\beta 4\text{GalT1}^{-/-}$ MKs, further data and more explanation are needed to explain the big difference between MKs in the BM and platelets in PB.

We apologize for not discussing the complexity of platelet production more in detail. The underlying pathogenic mechanisms of thrombocytopenia can be categorized as: (1) defects in MK lineage commitment and differentiation, (2) defects in MK maturation, and (3) defect in platelet release (also termed pro-platelet release or thrombopoiesis), i.e. extension of cytoplasmic protrusions into the blood stream. The late stages of thrombopoiesis require the formation of the so-called demarcation membrane system (DMS) for platelet release. The mechanisms leading to formation of the DMS and platelet release are not well understood. A paragraph has now been added to the introduction (p. 1, line 3-9).

Our data show that MK lineage commitment and MK maturation are normal in the absence of $\beta 4\text{GalT1}$ glycan synthesis.

By contrast, the process of thrombopoiesis (pro-platelet release into the blood stream) is highly affected *in vivo* and *in vitro*. Lack $\beta 4\text{GalT1}$ glycan synthesis renders $\beta 1$ -integrin hyperactive (see Figure 5 and 6), thus negatively affecting 1) megakaryocyte migration to bone marrow sinusoids and 2) organization of the demarcation membrane system, an essential precursor structure for proplatelet formation before their release into the blood stream (Figure 4).

Thus, our data provide evidence that $\beta 1$ integrin regulates this process by binding to various extracellular matrix components in the bone marrow, thus providing novel insight into this complex process. The data provide a clear explanation for low circulating platelets in the presence of high $\beta 4\text{GalT1}^{-/-}$ megakaryocyte numbers with impaired late stage platelet release.

Moreover, increased MK numbers often accompany thrombocytopenia and do not always predict platelet numbers or guarantee platelet release (thrombopoiesis). Additionally, during stress hematopoiesis, i.e. lack of platelet production as shown in this paper, platelet biased HSCs numbers increase, thus further contributing to the observed increase in megakaryocyte numbers.

2. In Figure 3F, $\beta 4\text{GalT1}^{-/-}$ BM cells could populate myeloid-lineage cells and MKs, but not B and T cells in the recipient BM after non-competitive transplants. However, $\beta 4\text{GalT1}^{-/-}$ BM cells showed normal ratio of B and T cells in Figure 2A, although T cells were slightly decreased. Please explain this discrepancy. Homing and engraftment of HSCs might be disturbed.

As expected, HSCs isolated from $\beta 4\text{GalT1}^{-/-}$ mice have a homing defects as referenced on p. 3 and below in ¹. HSCs are functionally heterogeneous ^{2,3}.

Transplantation of single HSCs has revealed reproducible bias toward selective differentiation to the myeloid or the lymphoid lineage ⁴. Using the von Willebrand factor promoter to drive GFP in transgenic mice (Vwf-eGFP), GFP⁺ was shown to mark a subset of quiescent HSCs that exhibit platelet and myeloid bias upon transplantation, whereas the GFP⁻ HSCs were lymphoid biased ⁵. Expression of c-Kit ⁶ or CD41 ⁷, used in the here shown study, also marks a megakaryocyte-biased HSC subset. This population increases under stress conditions, as reported by others.

Our data shows that at steady state thrombocytopenic $\beta 4\text{GalT1}^{-/-}$ mice contain more LT-HSCs that express CD41 (platelet specific marker) on their surface, further suggesting HSC platelet bias. This platelet-biased population is expected to increase under stress condition such as bone marrow transplantation. The focus of this manuscript is on thrombopoiesis and MKs. Our data shows that $\beta 4\text{GalT1}^{-/-}$ HSCs differentiate and mature into MKs after transplantation and produce increased numbers of MKs, suggesting HSC bias. Hence, MKs mature and differentiate but fail to produce platelets due to a thrombopoiesis defect.

To support that wild type MKs produce platelets normally, we now added total platelet count to Figure 3, showing normal platelet recoveries in the transplanted mice. These results support the notion that transplanted control megakaryocytes produce platelets normally (Figure 3).

3. In non-competitive transplants, ratio of GFP+ $\beta 4\text{GalT1}^{-/-}$ MKs at sinusoid was about half in Figure 3J, while ratio of GFP+ $\beta 4\text{GalT1}^{-/-}$ MKs in the whole BM increased 4-fold in Figure 3G, suggesting that the number of GFP+ $\beta 4\text{GalT1}^{-/-}$ MKs at sinusoid was 2-fold higher. However, GFP+ $\beta 4\text{GalT1}^{-/-}$ platelets in PB was less than 10%. Please explain why GFP+ $\beta 4\text{GalT1}^{-/-}$ platelets profoundly decreased, as asked in 1).

We apologize for not clarifying the platelet production defect. Our data clearly show that megakaryocytes differentiate and mature in the absence of $\beta 4\text{GalT1}$, but fail to release platelets, because of malformed DMS and impaired location at the sinusoids. See also point 1. This point has been now elaborated in the introduction p. 3 and p11.

4. $\beta 1$ -integrin was activated in $\beta 4\text{GalT1}^{-/-}$ MKs in Figure 5A-E. However, there was no direct evidence that active $\beta 1$ -integrin was agactosylated. Please compare the activity of fully galactosylated and agactosylated $\beta 1$ -integrin

We thank the Reviewer for the comment. As suggested, we immunoprecipitated $\beta 1$ integrin from WT and $\beta 4\text{GalT1}^{-/-}$ platelets and subjected $\beta 1$ integrin immunoprecipitates and total platelet lysates to SDS-PAGE and immunoblotting using Erythrina cristagalli lectin (ECL). The data show that $\beta 1$ integrin bears terminal galactose structures in WT platelets as determined by ECL binding, whereas terminal galactose was not detectable in $\beta 1$ integrin immunoprecipitated from $\beta 4\text{GalT1}^{-/-}$ platelets, suggesting that $\beta 1$ integrin does not bear glycans with terminal galactose residues in the absence of $\beta 4\text{GalT1}$. The data were corroborated using RCA 1 (not shown). The data are shown in Figure 5 and the results are described on p. 9, line 22 to 27. We now added data showing adhesion of control and $\beta 4\text{GalT1}^{-/-}$ platelets to $\beta 1$ integrin ligands in Figure 6, p.10. The data show increased $\beta 4\text{GalT1}^{-/-}$ platelet binding to the $\beta 1$ integrin ligands fibronectin, laminin and collagen, and normal binding to the $\beta 3$ integrin ligand fibrinogen. The finding suggests that the function of $\beta 3$ integrin is less affected by lack of galactosylation, the primary fibrinogen binding integrin in the platelet lineage.

5. In Figure 6B, GalT activity was not detected in $\beta 4\text{GalT1}^{-/-}$ MKs and platelets at all. However, MFIs of ECL signals were about half and 1/4 in $\beta 4\text{GalT1}^{-/-}$ MKs and platelets, respectively in Figure 6D. Please explain this discrepancy.

We agree with the Reviewer. The assay is specific for galactosyltransferase (GalT) activity measurement. The data shown in Figure 6 D and E were obtained by flow cytometry using Erythrina cristagalli lectin (ECL). Lectin binding by flow cytometry is associated with some background, which is not adjustable for due to lack of appropriate "isotype" controls. We carefully adjusted lectin concentrations to the lowest, not agglutinating levels of lectins. Agglutination of cells was ruled out by flow cytometry and direct microscopic visualization (not shown). Although ECL is widely used as a reagent for the detection of terminal galactopyranose (Gal) residues in glycans (its canonical specificity is for Gal), it also binds to N-acetylgalactosamine (GalNAc) and fucosylated

Gal (Fuc α 1–2Gal)⁸. Hence, we reason that the residual signal is due to the additional lectin binding specificities. This statement has now been added to the text on p.12.

6. ECL signals markedly increased in MKs by TPO and CXCL12 in Figure 6G, while β 4GalT1 mRNA levels slightly increased (1.3-fold) by TPO and CXCL12 in Figure 6H. Can the slight increase of β 4GalT1 mRNA account for the markedly increase of ECL signals?

We thank the Reviewer for the comment. The control mRNA levels were measured in the presence of thrombopoietin (TPO), which substantially increases ECL binding. Addition of CXCL12 to cultured megakaryocytes additionally increases β 4GalT1 mRNA expression, which is reflected in additional ECL binding. The native multi-valency of lectins may promote high-affinity avidity binding to the glycoproteins containing galactose glycans that are added by additional GalT activity and thereby add to increased ECL signal, p. 13.

7. ECL signals were examined on the surface and in whole cell lysates of MKs in Figure 6, but not on β 1-integrin. ECL signals on β 1-integrin should be examined.

We absolutely agree with the Reviewer. In Figure 5C, we now added experiments showing binding of ECL to immunoprecipitated β 1 integrin subjected to SDS-PAGE and immunoblotting. The data show, that ECL does not bind to β 1 integrin in the absence of β 4GalT1.

8. Global examinations about β 4GalT1^{-/-} mice, such as birth rate, weaning rate and body weight were also described in a previous paper (EMBO J 16: 1850-57, 1997). Please refer to this paper and discuss.

We apologize for the oversight. In our studies we used the mouse model generated by Lu et al⁹ and compare this model with the mouse model lacking β 4GalT1 generated by Asano et al^{1,10,11}. The discrepancies are now discussed on page p. 14.

9. Please show whole gel images of Figure 3A, 5A and 5B as supplemental figures. Whole gel images have been added to supplemental figures.

10. Since PF4 signals differed between control and β 4GalT1^{-/-} MK releasates, TGF- β /PF4 values are not appropriate in Figure 3D.

According to suggestions of multiple Reviewers, the data regarding primary myelofibrosis, TGF- β secretion and JAK signaling were perceived as too preliminary and were removed from this manuscript.

11. In Figure 6G, it appears that control gel was strongly exposed, while β 4GalT1^{-/-} gel was weakly exposed, because background signals were different. Both samples should be loaded in the same gel and exposed at the same level.

While we agree with the Reviewer, the immunoblot has been removed because these results were part of the primary myelofibrosis data.

12. In the text, Figure S2 is missing and Figure S3, which is not attached, is described.

We apologize for the oversight. The main text and figure legend have been corrected.

13. Figure S2 and its legend are incompatible.

See point 11.

14. In Experimental procedures, immunoblotting, before ECL lectin blotting, the membrane must be treated with sialidase to remove sialic acids. Without sialidase treatment, ECL cannot bind to galactosyl residues.

Our experiments aim to detect differences in the polypeptides that bear terminal galactose (asialylated glycoproteins) at steady state or under the pertinent experimental conditions, and do not aim to detect all galactose bearing glycoproteins. Sialidase treatment of immunoblots would hydrolyze all sialylated glycan residues. Hence, pre-treatment with sialidase will diminish/abolish differences in terminal galactose bearing glycoproteins under steady state conditions and following agonist stimulation.

15. On page 9, line 22, “two additional polypeptides” should be “three additional polypeptides”.

The sentence has been changed.

16. On page 13, line 2, “increased control” might be “increased more than control”.

The sentence was removed.

17. On page 27, title of Figure 2 legend, “ β 4GalT1^{-/-}” should be “ β 4GalT1”.

The title has been changed.

18. On page 28, title of Figure 4 legend, “ β 4GALT1” should be “ β 4GalT1”.

The title has been changed.

19. In Figure 8, “Platelet-biased HSC” might be in the right β 4GalT1^{-/-} model

The model has been changed.

REVIEWER #3

In this study the authors have studied the role of beta4GALT1 on hematopoiesis using beta4GALT1 knock out mice. The authors have focused their study on the HSC compartment and megakaryopoiesis as these mice have a profound thrombocytopenia and that the role of beta4GALT on neutrophils and the erythroid lineage have been previously studied.

General comment

1) There is a strong central message which is beta4GALT1 induces a defect in terminal megakaryocyte differentiation associated with a poor development of DMS leading to a profound defect in proplatelet formation. This defect is related to an activation of beta 1 integrin as a consequence of a defect in glycosylation.

The other parts of the papers are not so well demonstrated and in some way are weakening the central message. Thus in my opinion the paper must be reorganized and the mechanism by which beta 1 activation impairs megakaryocyte differentiation must be further studied while the other parts could be omitted.

Specific comment:

2) A significant part of the paper is dedicated to the hematopoietic stem cell compartment and is only based on the immunophenotype of the HSC compartment. In fact to characterize this compartment and the presence of a megakaryocyte/platelet bias it will be necessary to perform ScRNA-seq and transplantations. However transplantation in this mixed genetic background is difficult and the transplantations in NSG mice are not the best manner to perform a detailed analysis. Furthermore as half of the homozygous knock out mice died, we do not know if a selection occurs which could be hematopoietic. Therefore such studies will require to have a new model of mice i.e. an inducible knockout.

We completely agree with all of the Reviewers' comments. According to the suggestions we now refocused the paper more on thrombopoiesis, removed the primary myelofibrosis data and added more HSC transplant data. Unfortunately, at this moment conditional or inducible null mice are not available.

To circumvent the lack of conditional mice we performed HSC transplants to verify that the defect in thrombopoiesis is intrinsic to MK and not dependent on environmental cues. We agree that transplants into NSG mice are difficult and added therefore transplant data using HSCs isolated from fetuses at E14.5 on the C57BL/6J background. Of note, at E14.5 mice are viable and have the expected mendelian ratio, therefore an HSC bias is unlikely at this point. Cells isolated from fetuses were transplanted into lethally irradiated C56BL/6J mice including supporting cells and resulted in profound thrombocytopenia when β 4GalT1^{-/-} cells were transplanted similar to the phenotype observed in NSG transplanted mice. Data were added to Figure 3 and p. 6-7. Our data shows that HSCs lacking β 4GalT1 differentiate into mature MKs that fail to release platelets. Our data also show that LT-HSCs have increased CD41 surface expression, indicative of platelet bias and transplants result in increased number of MKs. To further substantiate the intrinsic defect in thrombopoiesis, we generate mice that specifically lack the β 1 integrin in MKs in β 4GalT1^{-/-} mice, a maneuver that significantly improved thrombopoiesis. This data is specific for MKs and thus supports the intrinsic MK

defect. This paper is now more focused on thrombopoiesis. We thus removed the term platelet biased HSCs and replaced it by perturbed HSCs.

3) In the text it is not described what are the control mice in this mixed genetic background? Is it wild type littermate?

We apologize for not being clear. Yes, the control mice were all wild type littermates. This information has now been added to the methodology on p. 18.

4) In addition as the expression of CD41 on HSC greatly increases with aging, at what age was performed the studies and had the knock out and control mice the same age?

We thank the Reviewer for pointing this out. We are fully aware of the age dependence of platelet-biased HSCs. In all experiments, HSC platelet bias and CD41 expression in HSCs was performed in 9-12 weeks old mice, with an equal distribution of male and female mice to exclude a sex bias of the phenotype. We used wild type littermates as controls, hence the age and genetic background used in each experiment were the same.

5) Why the authors did not study the spleen as in the initial description of the mice an extramedullary hematopoiesis was present?

We thank the Reviewer for the comment. We focused our investigation on bone marrow hematopoiesis and bone marrow megakaryocytes. While the spleen has megakaryocytes, these megakaryocytes appear to have very little contribution to hematopoiesis at steady state, thus the data were not included as a signature of extramedullary hematopoiesis of primary myelofibrosis. In table 2 we now added data for LT-HSC, ST-HSC, and MPP numbers in the spleen in table 2 and text on p. 6. The data show that LT-HSCs and MPPs are increased (not significantly), suggesting extramedullary hematopoiesis.

6) The part on the TGF beta is partially convincing. Why have the authors studied the TGF beta level by western blots in megakaryocyte lysates and in the megakaryocyte supernatants? I think that an Elisa would have been more quantitative and could have also permitted to discriminate between the active and inactive form. It is unclear why there is an active form already in the MK lysate. It is usually considered the TGF beta 1 is stored under an inactive form and activated after secretion. Most of the studies have been performed on in vitro cultured fetal megakaryocytes, which could be an important bias in comparison to the bone marrow. Why the level of TGF beta has not been studied in the plasma and the spleen? The manner to quantify TGF beta in the culture supernatant is also difficult to understand. This is done in comparison to PF4. However in the text the authors assess that the level of secreted PF4 is increased for beta4GALT1 minus megakaryocytes. Finally the ratio between the active and inactive form in beta4GALT1 mice is not modified, thus there is no defect in TGF beta1 activation, but may be a defect in synthesis. Did the authors study the level of TGF beta1 transcript in megakaryocytes?

We agree with the Reviewer that the data are preliminary and require more investigation in conjunction with primary myelofibrosis. We also thank the Reviewer for the great suggestions to improve the experiments. However, the data have been removed and will be published elsewhere in a later publication. We refocused this paper on thrombopoiesis and $\beta 1$ integrin function. As pointed out to Reviewer #1 we generated two state-of-the-art mouse models of PMF obtained by transplanting HSCs over-expressing thrombopoietin and by chronic administration of the thrombopoietin receptor agonist Romiplostim. Despite a clear increase in terminal galactosylation by several cell types and extracellular matrix components following the development of myelofibrosis, both mouse models did not phenocopy the megakaryocyte phenotype found in patients with PMF, hence hampering our investigation. The data require more investigation and will be included in a future manuscript.

7) The third part on beta4GalT1 and megakaryocyte differentiation is well performed and convincing for the role of beta1 integrin activation. However the precise mechanism remains to be understood. How beta 1 integrin activation impairs the development of DMS? Are some Rho GTPases implicated in this inhibition of megakaryopoiesis? It will be interesting to study in more details how these knock out megakaryocytes adhere to different matrix such as collagen or fibronectin and what are the changes in the cytoskeleton as well as in cell migration.

We thank the Reviewer for the positive comment. We agree that the mechanism how $\beta 1$ integrin function affects DMS formation is an important point of investigation. We here focused on the role of posttranslational

modifications in the regulation $\beta 1$ integrin function in MKs and platelets and upstream signaling. The downstream events of $\beta 1$ integrin activation are beyond the scope of this paper. We now added data that unequivocally show that $\beta 4\text{GalT}1$ is required to add galactose to $\beta 1$ integrin in MKs. MKs express $\alpha 2$, $\alpha 4$, $\alpha 5$ and $\alpha 6$ subunits, that can form complexes with $\beta 1$ integrin, and αIIb as part of the platelet specific fibrinogen receptor $\alpha \text{IIb}\beta 3$. We here determined the apparent molecular weight of $\alpha 5$ and $\alpha 6$ subunits by immunoblotting and show that all the molecular weight of all subunits differs from control in the absence of $\beta 4\text{GalT}1$. 3) $\beta 4\text{GalT}1^{-/-}$ platelets adhere more avidly to laminin, collagen and fibronectin ($\beta 1$ integrin ligands) compared to control, whereas adhesion to fibrinogen ($\alpha \text{IIb}\beta 3$ integrin ligand) was comparable to control (Figures 5 and 6) and p 8-10. These data suggest that multiple $\beta 1$ integrin complexes and interactions with extracellular regulate DMS formation and platelet release. We also show that FAK is phosphorylated thus the data collectively suggest cytoskeletal changes occurring during the platelet release process.

8) How do the authors explain that platelets have a normal half live? Is the level of TPO normal in the plasma not only in bone marrow supernatant?

Published data point to the role of $\text{GPIb}\alpha$ as a clearance receptor. $\beta 4\text{GalT}1$ adds galactose mainly in N-linked glycans and mouse $\text{GPIb}\alpha$, although extensively O-glycosylated, does not bear N-glycans. The apparent molecular weight and function of $\text{GPIb}\alpha$ in $\beta 4\text{GalT}1^{-/-}$ platelets are comparable to control, suggesting a minimal or no effect of lack of $\beta 4\text{GalT}1$ in posttranslational regulation of this molecule in platelets. Hence, it is possible that lectins do not recognize the minimal change to clear platelets. We can only speculate that the terminal GlcNAc epitopes are not recognized by $\alpha \text{M}\beta 2$ integrin. We now added plasma TPO levels in Table 2, which are ~4 times higher in $\beta 4\text{GalT}1^{-/-}$ mice compared to control.

9) The fourth part on the regulation by TPO and CXCL12 is also not convincing for several reasons. in Figure 6 H there is not the level of beta4GALT1 in the absence of TPO or in the presence of CXCL12 alone. In addition AZD 1480 is used at high concentration and at this concentration it does not only inhibit JAK1/JAK2 but also Aurora Kinases. Thus lower doses and the use of other inhibitors are required. Did the authors stimulate megakaryocyte by IL6 to induce the JAK/STAT pathway? As CXCL12 has no effect alone, the absence of effect of CXCL12 with TPO in the presence of a JAK inhibitor does not demonstrate that CXCL12 requires JAK2.

The data were removed from this manuscript.

10) Did the authors study the level of beta4GalT1 in c-mpl constitutive knock out megakaryocytes, which will be a manner to know if beta4GALT1 is regulated by TPO? If their hypothesis is true, how the authors explain that the deletion of c-mpl or Jak2 specifically in megakaryocytes with the PF4-Cre induce an important thrombocytosis? If they were right it is expected a profound thrombocytopenia.

All of these mouse models are complex and require more investigation. The data has been removed from this paper.

11) The work on patients with PMF is a totally different topic. Indeed in this part the authors show that in these patients with a constitutive JAK2 activation the Beta4GALT1 is increased and in some way correlates with the level of fibrosis. So the authors have first to show that an increase in Beta4 GALT1 level using transgenic mice or lentiviral vector will induce a myelofibrosis or at least an increase of active TGF beta1 release before to study patients.

We generated two state-of-the-art mouse models of PMF obtained by transplanting HSCs over-expressing thrombopoietin and by chronic administration of the thrombopoietin receptor agonist Romiplostim. Despite a clear increase in terminal galactosylation by several cell types and extracellular matrix components following the development of myelofibrosis, both mouse models did not phenocopy the megakaryocyte phenotype found in patients with PMF, hence hampering our investigation. This investigation will require more work and will be published elsewhere.

In addition 6 of the patients described in Table 3 have not a PMF because they have either no fibrosis (1 patient) and 5 a grade 1. Using the WHO classification, at least a grade 2 fibrosis is required for the diagnosis of PMF. I think that this part is not at all relevant for this paper.

We agree with the Reviewer. The patient data have been removed and will be published at another time.

Many references are reviews, it will be much better to add the specific references.
Reviews have been replaced by original papers.

REVIEWER #4

1. What is the role(s) of integrin $\beta 1$ on the CXCL12 (SDF1), FGF1, and TGF β cytokine signaling?
The data concerning cytokine secretion have been perceived by all Reviewers as too preliminary and have been therefore removed from this paper.

2. There is no mechanism of why $\beta 4$ GalT1 knockout mature staged MKs are NOT located at sinusoid site. How $\beta 1$ integrin with LacNAc (glycosylated structure) regulated the location of MKs?

We apologize for not being clear. Here we demonstrate: 1) Lack of $\beta 4$ GalT1 leads to agalactosylation and thus asialylation of $\beta 1$ integrin. We now provide clear evidence that $\beta 1$ integrin is agalactosylated in Figure 5. 2) SDF1 promotes migration of megakaryocyte progenitors to sinusoids¹². Our data suggests that SDF1 increases galactosylation of megakaryocyte glycoproteins including $\beta 1$ integrin a prerequisite for sialic acid substitutions. 3) We provide evidence that $\beta 4$ GalT1^{-/-} platelets bind more avidly to laminin, collagen and fibronectin compared to control (Figure 6). Thus, our data suggest that increased adhesion to ECM in the bone marrow could slow megakaryocyte progenitor migration and that SDF1 promotes additional galactosylation and additional sialylation of $\beta 1$ integrin to promote megakaryocyte progenitor migration towards sinusoids.

Gain of function mutations in genes encoding for α IIb and $\beta 3$ result in constitutively active α IIb $\beta 3$ and are associated with thrombocytopenia in patients. Megakaryocytes differentiated from CD34+ cells from two of the patients with constitutively active α IIb $\beta 3$ were studied *in vitro*¹³. While megakaryocyte maturation and differentiation did not significantly differ between patients and controls, proplatelets formed by patient's megakaryocytes were significantly reduced leading to thrombocytopenia. Megakaryocytes plated on fibrinogen were spreading and formed shorter and lower numbers of proplatelet shafts compared to controls¹³. These data show that constitutive activation of α IIb $\beta 3$ causes thrombocytopenia due to a defect in proplatelet production and platelet release, similar to the here described phenotype.

Mature megakaryocytes localize at bone marrow sinusoids. CXCL12 (SDF-1) is as a potent chemoattractant for MKs, leading to their migration towards the sinusoids in a $\alpha 4\beta 1$ /VCAM dependent manner thereby increasing platelet counts in MPL (TPO receptor) deficient mice¹². Although mature MKs reside at the vessels, very slow migration of MKs was detected with no differences between young and mature MKs and no changes in their localization following CXCL12 treatment or CXCR4 (CXCL12 receptor) blockage. These data suggest a revised model of thrombopoiesis where MKs reside at the sinusoids and suggest that hematopoietic progenitors migrate towards vessels where they complete their full differentiation into MKs. MK migration has been studied *in vitro*¹⁴⁻¹⁶, however the lack of a bone marrow environment and signals hampers the interpretation of *in vitro* data.

Our data provides novel insight to MK progenitor migration/localization *in vivo* in mice showing 1) that hyperactive $\beta 1$ integrin affects localization and 2) posttranslational modification regulate $\beta 1$ integrin function in MKs *in vivo*.

3. What is the counter-subunit α of $\beta 1$ integrin? $\alpha 2\beta 1$ for collagen I, $\alpha 4\beta 1$ for VCAM-1/Fibronectin, $\alpha 6\beta 1$ for laminin? If all integrins are involved in this story, the data should provide the cases of individual ECM and specified integrin cascade for this story.

We determined the apparent molecular weight of $\alpha 5$ and $\alpha 6$ subunits by immunoblotting and show that all the molecular weight of all subunits differs from control in the absence of $\beta 4$ GalT1. Furthermore, $\beta 4$ GalT1^{-/-} platelets adhere more avidly to laminin, collagen and fibronectin ($\beta 1$ integrin ligands) compared to control, whereas adhesion to fibrinogen (α IIb $\beta 3$ ligand) was comparable to control (Figures 5 and 6) and p 8-9. In agreement with the reviewers' comments these data suggest that multiple $\beta 1$ integrin complexes and interactions with extracellular matrix components regulate DMS formation and platelet release (see also

reviewer 1). By contrast, the function of α IIb β 3 is normal (Figure 5 and 6), suggesting that posttranslational modifications mediated by β 4GalT1 affect β 1 integrin function specifically and do not affect α IIb β 3. Because multiple β 1 integrins play a role in DMS formation and thrombopoiesis, we deleted this subunit in megakaryocytes a maneuver which allowed us investigate the role of *all* β 1 integrins in megakaryocytes.

We provide evidence that protein tyrosine phosphorylation, particularly of FAK, are increased in the presence of agalactosylated β 1 integrin, and deletion of β 1 integrins normalizes FAK tyrosine phosphorylation. These data suggest that all galactosylation of β 1 integrins regulates platelet release and migration likely via cytoskeletal elements. However, the likely followed by sialic acid substitutions investigation of the individual signaling pathways of each integrin is beyond the scope of this paper.

4. Why β 1-integrin subunit deletion significantly improved circulating platelet count to 70%?

Deletion of β 1-integrin in megakaryocytes has no effect on circulating platelet count^{17,18}. Similarly, deletion of α IIb β 3 integrin (such as in Glanzman Thrombasthenia) does not affect platelet count, i.e thrombopoiesis. Thus, loss of normal integrin functions can be compensated by other mechanisms in megakaryocytes. By contrast hyperactive α IIb β 3 integrin inhibits thrombopoiesis¹⁹⁻²¹, suggesting that hyperactive integrin function cannot be overcome by other mechanisms. Deletion of the *hyperactive* β 1 integrin *specifically in MKs* normalized platelet production and platelet count to 70% showing that hyperactive integrin are not compatible with normal thrombopoiesis and have to their activity has to be carefully regulated.

5. It is pretty difficult to understand the following statement, “Together, the data shows that absence of β 4GalT1 does not affect HSC quiescence but promotes HSC numbers and platelet-bias due to impaired thrombopoiesis and secretion of cytokines from β 4GalT1^{-/-} MKs.” Because in MK-HSC niche system diminished TGF β levels might induce the escape from dormancy of HSC leading to cell cycle acceleration at HSC/MPP levels. Why did you conclude that absence of β 4GalT1 does not affect HSC quiescence? There are lacks of information regarding interaction between several cytokine influences in this manuscript. This is weak point.

We agree with the Reviewer and rephrased this sentence on p. 6.

6. Another weak point is that the author did not use conditional β 4GalT1 knockout mice specifically targeting megakaryocyte to directly evaluate the effect of β 4GalT1 knockout on platelet release from MKs in adult mice. We completely agree with the Reviewer. Unfortunately, conditional knockouts are currently not available. We used two approaches to circumvent this weakness. 1) To exclude that extracellular components, specifically extracellular matrix components, affect platelet production in the absence β 4GalT1 we performed HSC transplants (Figure 3) and in vitro experiments (Figure 4). 2) Deletion of active β 1-integrin specifically in megakaryocytes using PF4-Cre clearly demonstrated that hyperactive β 1 integrin in megakaryocytes significantly improved thrombopoiesis providing specific megakaryocyte related evidence that absence of β 4GalT1 affects β 1-integrin function and thrombopoiesis.

7. The glycosylation of β 1-integrin by β 4GalT1 is required for platelet release through suppression of integrin active form, but not megakaryocyte maturation? Or, stable β 4GalT1 structure somehow influence the inhibition of active form of β 1 integrin?

We suggest that the glycosylation mediated by β 4GalT1 in the β 1 subunit and likely also in subunits associated with β 1 decreases/modifies their ligand binding, thereby facilitating MK localization and platelet release.

8. In this case, inside-out signaling or direct outside-in signaling of integrin is involved?

To investigate β 1 integrin function, we performed platelet adhesion studies using β 1 integrin ligands (fibronectin, collagen, and laminin). Our data show that β 4GalT1^{-/-} platelets increased more avidly to all β 1 integrin ligands compared to control (figure 6). By contrast binding of β 4GalT1^{-/-} platelets to fibrinogen was normal, suggesting that lack of galactosylation specifically affects β 1 integrins and has a lesser effect on the α IIb β 3, the major fibrinogen binding receptor complex in MKs and platelets.

9. This reviewer is also curious in general that glycosyltransferases is the most critical switch of platelet shedding initiation from MKs?

We apologize for not discussing the complexity of platelet production more in detail. The underlying pathogenic mechanisms of thrombocytopenia can be categorized as (1) defects in MK lineage commitment and differentiation, (2) defects in MK maturation, and (3) defect in platelet release (also termed pro-platelet release or thrombopoiesis) i.e. extension of cytoplasmic protrusions into the blood stream. The late stages of thrombopoiesis require the formation of the so-called demarcation membrane system (DMS) for platelet release. The mechanisms leading to formation of the DMS and platelet release are not well understood. A paragraph has now been added to the Introduction (p. 3)

The role of glycosylation becomes increasingly evident in thrombopoiesis. However, most reports provide specific genetic alterations but do not provide mechanistic insight. Our findings provide novel mechanistic insight into the role of glycosylation in thrombopoiesis. Our data shows that MK lineage commitment and MK maturation are normal in the absence of $\beta 4\text{GalT-1}$ glycan synthesis. By contrast, the process of thrombopoiesis (pro-platelet release into the blood stream) is highly affected *in vivo* and *in vitro*. Lack of $\beta 4\text{GalT-1}$ glycan synthesis renders $\beta 1$ -integrin hyperactive (see Figure 5 and 6) thus negatively affecting 1) megakaryocyte migration to bone marrow sinusoids and 2) organization of the demarcation membrane system, an essential precursor structure for proplatelet formation before their release into the blood stream (Figure 4).

10. It is assumed that glycosylation takes place for protein biosynthesis at Golgi apparatus. How glycosylation is completed in $\beta 1$ integrin structure at Golgi?

We thank the Reviewer for the interesting question. Very little is known about Golgi and ER function in MKs. Studies of Golgi and MK function are complex in the multinucleated cell. Hence, it will require substantial amounts of work to precisely identify the localization of $\beta 1$ integrin modifications in MKs, which is beyond the scope of this manuscript. This interesting question will be addressed in the future.

11. There are any differences of PLT functionalities between $\beta 4\text{GalT1 KO}$ and WT PLTs?

$\beta 4\text{GalT1}^{-/-}$ platelet function data have been added in Figure 6. We have evaluated α -granule release (P-selectin surface exposure) and activation of the fibrinogen receptor $\alpha \text{IIb}\beta 3$ following thrombin activation *in vitro* with no measurable differences between the phenotypes. Von Willebrand Factor receptor (GPIIb/IIIa) function was evaluated by ristocetin-induced von Willebrand factor (vWf) binding to platelets and showed no difference in vWf binding between the phenotypes. To specifically investigate $\beta 1$ integrin function, we performed platelet adhesion studies using $\beta 1$ -integrin ligands (fibronectin, collagen and laminin) *in vitro* (see also Reviewer #1 and #4). The data clearly show that $\beta 4\text{GalT1}^{-/-}$ platelets adhere more avidly to all of the tested $\beta 1$ integrin ligands further supporting notion that $\beta 1$ integrin is hyperactive in the absence of $\beta 4\text{GalT1}$ mediated glycan synthesis. Additionally, we have now provided evidence that $\beta 1$ integrin is agalactosylated in the absence of $\beta 4\text{GalT1}$ (Figure 6).

References

1. Takagaki, S., *et al.* Galactosyl carbohydrate residues on hematopoietic stem/progenitor cells are essential for homing and engraftment to the bone marrow. *Scientific reports* **9**, 7133 (2019).
2. Benz, C., *et al.* Hematopoietic stem cell subtypes expand differentially during development and display distinct lymphopoietic programs. *Cell Stem Cell* **10**, 273-283 (2012).
3. Muller-Sieburg, C.E., Sieburg, H.B., Bernitz, J.M. & Cattarossi, G. Stem cell heterogeneity: implications for aging and regenerative medicine. *Blood* **119**, 3900-3907 (2012).
4. Dykstra, B., *et al.* Long-term propagation of distinct hematopoietic differentiation programs *in vivo*. *Cell Stem Cell* **1**, 218-229 (2007).
5. Sanjuan-Pla, A., *et al.* Platelet-biased stem cells reside at the apex of the haematopoietic stem-cell hierarchy. *Nature* **502**, 232-236 (2013).
6. Shin, J.Y., Hu, W., Naramura, M. & Park, C.Y. High c-Kit expression identifies hematopoietic stem cells with impaired self-renewal and megakaryocytic bias. *J Exp Med* **211**, 217-231 (2014).
7. Gekas, C. & Graf, T. CD41 expression marks myeloid-biased adult hematopoietic stem cells and increases with age. *Blood* **121**, 4463-4472 (2013).

8. Manimala, J.C., Roach, T.A., Li, Z. & Gildersleeve, J.C. High-throughput carbohydrate microarray analysis of 24 lectins. *Angew Chem Int Ed Engl* **45**, 3607-3610 (2006).
9. Lu, Q., Hasty, P. & Shur, B.D. Targeted mutation in beta1,4-galactosyltransferase leads to pituitary insufficiency and neonatal lethality. *Dev Biol* **181**, 257-267 (1997).
10. Asano, M., *et al.* Growth retardation and early death of beta-1,4-galactosyltransferase knockout mice with augmented proliferation and abnormal differentiation of epithelial cells. *EMBO J* **16**, 1850-1857 (1997).
11. Asano, M., *et al.* Impaired selectin-ligand biosynthesis and reduced inflammatory responses in beta-1,4-galactosyltransferase-I-deficient mice. *Blood* **102**, 1678-1685 (2003).
12. Avezilla, S.T., *et al.* Chemokine-mediated interaction of hematopoietic progenitors with the bone marrow vascular niche is required for thrombopoiesis. *Nat Med* **10**, 64-71 (2004).
13. Bury, L., Malara, A., Gresele, P. & Balduini, A. Outside-in signalling generated by a constitutively activated integrin alphaIIb beta3 impairs proplatelet formation in human megakaryocytes. *PLoS One* **7**, e34449 (2012).
14. Hamada, T., *et al.* Transendothelial migration of megakaryocytes in response to stromal cell-derived factor 1 (SDF-1) enhances platelet formation. *J Exp Med* **188**, 539-548 (1998).
15. Dhanjal, T.S., *et al.* A novel role for PECAM-1 in megakaryocytokinesis and recovery of platelet counts in thrombocytopenic mice. *Blood* **109**, 4237-4244 (2007).
16. Mazharian, A., Thomas, S.G., Dhanjal, T.S., Buckley, C.D. & Watson, S.P. Critical role of Src-Syk-PLC γ 2 signaling in megakaryocyte migration and thrombopoiesis. *Blood* **116**, 793-800 (2010).
17. Petzold, T., *et al.* beta1 integrin-mediated signals are required for platelet granule secretion and hemostasis in mouse. *Blood* **122**, 2723-2731 (2013).
18. Holtkotter, O., *et al.* Integrin alpha 2-deficient mice develop normally, are fertile, but display partially defective platelet interaction with collagen. *J Biol Chem* **277**, 10789-10794 (2002).
19. Bury, L., *et al.* Cytoskeletal perturbation leads to platelet dysfunction and thrombocytopenia in variant forms of Glanzmann thrombasthenia. *Haematologica* **101**, 46-56 (2016).
20. Kashiwagi, H., *et al.* Demonstration of novel gain-of-function mutations of alphaIIb beta3: association with macrothrombocytopenia and glanzmann thrombasthenia-like phenotype. *Mol Genet Genomic Med* **1**, 77-86 (2013).
21. Favier, M., *et al.* Mutations of the integrin alphaIIb/beta3 intracytoplasmic salt bridge cause macrothrombocytopenia and enlarged platelet alpha-granules. *Am J Hematol* **93**, 195-204 (2018).

REVIEWERS' COMMENTS:

Reviewer #1 (Remarks to the Author):

The authors have adequately addressed all of my previous concerns.

I only have a minor correction: there is a comment on the bottom (last line) of page 11, that should be removed.

Reviewer #2 (Remarks to the Author):

The authors have well rived the manuscript according to the comments of the reviewer #2. The reviewer #2 thinks the manuscript can be accepted after correcting minor points described below.

Answers to reviewer #2

8. Another beta4GalT-1-/- mouse generated by Asano et al. is described well in the manuscript. However, Asano et al did not mention about beta4GalT-1-/- mice on C57BL/6 genetic background at all. So there is no discrepancy about embryonic lethality of them on C57BL/6 background. Although the above paper by Asano et al. is referred as 43, referred 43 paper in the reference is not correct.

12, 13. In Figure S2 legend (B-C), "anti-PF4 mAb" is missing.

Reviewer #4 (Remarks to the Author):

I wonder why beta3 integrin is independent of glycosylation, quite different from the case of beta1 integrin. Why do beta1 integrins mechanistically require glycosylation process towards PLT biogenesis from megakaryocytes?

Regarding other points, I understood mostly. The revised version of manuscript and response letter are satisfactory.

Reviewer #5 (Remarks to the Author):

The comments made by reviewer 3 were carefully addressed by the authors. Data considered by this reviewer too preliminary, were deleted (concern 6,9,10 and 11) and the revised manuscript now focuses on the regulation of B1 integrin function by beta4GALT in megakaryocytes using suited mouse models. this message had been found meritorious by reviewer 3. Removal of these data does not decrease but rather enhances the clarity of the presentation by increasing focus.

The comments made by Reviewer 3 on sections which were retained by the authors were all addressed in a satisfactory fashion as indicated below:

General comment

1) There is a strong central message which is beta4GALT1 induces a defect in terminal megakaryocyte

differentiation associated with a poor development of DMS leading to a profound defect in proplatelet

formation. This defect is related to an activation of beta 1 integrin as a consequence of a defect in glycosylation.

As suggested by this reviewer, the authors do focus their paper on the central message pointed out with meritorious wording by reviewer 3.

Specific comment:

Concern 2: immunophenotyping alone is not sufficient to demonstrate a megakaryocyte/platelet bias of hematopoietic stem cells. The authors have addressed this concern by removing any discussion for a possible "megakaryocyte" bias of the mutant hematopoietic stem cells.

Immunophenotype studies were complemented by transplantation of hematopoietic stem cells isolated from the fetal liver in wild-type C57BL/6J background. This is an accepted approach to test the functions of hematopoietic stem cells from mice that die at birth (new data included in Figure 3). In addition, the authors supported their conclusion by rescuing the phenotype of $\beta 4\text{GalT1}^{-/-}$ mice

with megakaryocytic specific deletion of $\beta 1$ integrin. The new data strongly support the authors' conclusion.

Comment 5) Why the authors did not study the spleen as in the initial description of the mice an extramedullary

hematopoiesis was present?

The data on the spleen are included in table 2, as requested. They are effective in demonstrating the point.

Concern 7) The third part on beta4GalT1 and megakaryocyte differentiation is well performed and convincing for

the role of beta1 integrin activation. However the precise mechanism remains to be understood. How beta 1

integrin activation impairs the development of DMS? Are some Rho GTPases implicated in this inhibition of

megakaryopoiesis? It will be interesting to study in more details how these knock out megakaryocytes adhere

to different matrix such as collagen or fibronectin and what are the changes in the cytoskeleton as well as in

cell migration.

The authors rightly point out that the mechanism how $\beta 1$ integrin function affects DMS formation is outside the purpose of this manuscript that is the cross talk between beta4Gal1T1 and beta1 integrin in the regulation of megakaryocytopoiesis.

Minor comments

Minor concern 3) In the text it is not described what are the control mice in this mixed genetic background? Is it wild type

littermate? This comment is clearly addressed by the revised manuscript.

Minor concern 4) In addition as the expression of CD41 on HSC greatly increases with aging, at what age was performed

the studies and had the knock out and control mice the same age? Age and gender of the mice is clarified by the revised manuscript

Minor concern 8) How do the authors explain that platelets have a normal half live? Is the level of TPO normal in the

plasma not only in bone marrow supernatant?

I am confused by this comment because it is debatable whether platelet release is regulated by TPO. Data from the laboratory of Alexander Warren clearly indicate that TPO is dispensable for terminal megakaryocyte maturation.

Comment indicated as minor by the reviewer: many references are reviews, it will be much better to add the specific references.

The authors replaced the Reviews with original papers, as suggested.

REVIEWERS' COMMENTS:

Reviewer #1 (Remarks to the Author):

The authors have adequately addressed all of my previous concerns. I only have a minor correction: there is a comment on the bottom (last line) of page 11, that should be removed.

The sentence was removed.

Reviewer #2 (Remarks to the Author):

The authors have well rived the manuscript according to the comments of the reviewer #2. The reviewer #2 thinks the manuscript can be accepted after correcting minor points described below.

Answers to reviewer #2

8. Another beta4GalT-1-/- mouse generated by Asano et al. is described well in the manuscript. However, Asano et al did not mention about beta4GalT-1-/- mice on C57BL/6 genetic background at all. So there is no discrepancy about embryonic lethality of them on C57BL/6 background. Although the above paper by Asano et al. is referred as 43, referred 43 paper in the reference is not correct.

The text and reference has been corrected.

In Figure S2 legend (B-C), "anti-PF4 mAb" is missing.

The Figure legend has been corrected.

Reviewer #4 (Remarks to the Author):

I wonder why beta3 integrin is independent of glycosylation, quite different from the case of beta1 integrin. Why do beta1 integrins mechanistically require glycosylation process towards PLT biogenesis from megakaryocytes?

At this point in this manuscript we are not be able to address this important and interesting question. We hope to address the function of glycans for integrin function in another manuscript.

Regarding other points, I understood mostly. The revised version of manuscript and response letter are satisfactory.

Reviewer #5 (Remarks to the Author):

The comments made by reviewer 3 were carefully addressed by the authors. Data considered by this reviewer too preliminary, were deleted (concern 6,9,10 and 11) and the revised manuscript now focuses on the regulation of B1 integrin function by beta4GALT in megakaryocytes using suited mouse models. this message had been found meritorious by reviewer 3. Removal of these data does not decrease but rather enhances the clarity of the presentation by increasing focus.

The comments made by Reviewer 3 on sections which were retained by the authors were all addressed in a satisfactory fashion as indicated below:

General comment

1) There is a strong central message which is beta4GALT1 induces a defect in terminal megakaryocyte

differentiation associated with a poor development of DMS leading to a profound defect in proplatelet formation. This defect is related to an activation of beta 1 integrin as a consequence of a defect in glycosylation.

As suggested by this reviewer, the authors do focus their paper on the central message pointed out with meritorious wording by reviewer 3.

Specific comment:

Concern 2: immunophenotyping alone is not sufficient to demonstrate a megakaryocyte/platelet bias of hematopoietic stem cells. The authors have addressed this concern by removing any discussion for a possible "megakaryocyte" bias of the mutant hematopoietic stem cells. Immunophenotype studies were complemented by transplantation of hematopoietic stem cells isolated from the fetal liver in wild-type C57BL/6J background. This is an accepted approach to test the functions of hematopoietic stem cells from mice that die at birth (new data included in Figure 3). In addition, the authors supported their conclusion by rescuing the phenotype of $\beta 4\text{GalT1}^{-/-}$ mice with megakaryocyte specific deletion of $\beta 1$ integrin. The new data strongly support the authors' conclusion.

Comment 5) Why the authors did not study the spleen as in the initial description of the mice an extramedullary hematopoiesis was present?

The data on the spleen are included in table 2, as requested. They are effective in demonstrating the point.

Concern 7) The third part on beta4GalT1 and megakaryocyte differentiation is well performed and convincing for the role of beta1 integrin activation. However the precise mechanism remains to be understood. How beta 1 integrin activation impairs the development of DMS? Are some Rho GTPases implicated in this inhibition of megakaryopoiesis? It will be interesting to study in more details how these knock out megakaryocytes adhere to different matrix such as collagen or fibronectin and what are the changes in the cytoskeleton as well as in cell migration.

The authors rightly point out that the mechanism how $\beta 1$ integrin function affects DMS formation is outside the purpose of this manuscript that is the cross talk between beta4GalT1 and beta1 integrin in the regulation of megakaryocytopoiesis.

Minor comments

Minor concern 3) In the text it is not described what are the control mice in this mixed genetic background? Is it wild type littermate?

This comment is clearly addressed by the revised manuscript.

Minor concern 4) In addition as the expression of CD41 on HSC greatly increases with aging, at what age was performed

the studies and had the knock out and control mice the same age? Age and gender of the mice is clarified by the revised manuscript

Minor concern 8) How do the authors explain that platelets have a normal half life? Is the level of TPO normal in the plasma not only in bone marrow supernatant?

I am confused by this comment because it is debatable whether platelet release is regulated by TPO. Data from the laboratory of Alexander Warren clearly indicate that TPO is dispensable for terminal megakaryocyte maturation.

Thank you.